elinks

# Adult-born neurons modify excitatory synaptic transmission to existing neurons

Elena W Adlaf, Ryan J Vaden, Anastasia J Niver, Allison F Manuel, Vincent C Onyilo, Matheus T Araujo, Cristina V Dieni, Hai T Vo, Gwendalyn D King, Jacques I Wadiche*, Linda Overstreet-Wadiche*

Department of Neurobiology, University of Alabama at Birmingham, Birmingham, United States

**Abstract** Adult-born neurons are continually produced in the dentate gyrus but it is unclear whether synaptic integration of new neurons affects the pre-existing circuit. Here we investigated how manipulating neurogenesis in adult mice alters excitatory synaptic transmission to mature dentate neurons. Enhancing neurogenesis by conditional deletion of the pro-apoptotic gene *Bax* in stem cells reduced excitatory postsynaptic currents (EPSCs) and spine density in mature neurons, whereas genetic ablation of neurogenesis increased EPSCs in mature neurons. Unexpectedly, we found that *Bax* deletion in developing and mature dentate neurons increased EPSCs and prevented neurogenesis-induced synaptic suppression. Together these results show that neurogenesis modifies synaptic transmission to mature neurons in a manner consistent with a redistribution of pre-existing synapses to newly integrating neurons and that a non-apoptotic function of the Bax signaling pathway contributes to ongoing synaptic refinement within the dentate circuit.

*For correspondence: jwadiche@uab.edu (JIW); lwadiche@uab.edu (LO-W)

Competing interests: The authors declare that no competing interests exist.

## Introduction

Continual neurogenesis in the adult dentate gyrus (DG) produces new granule cells (GCs) that integrate into the hippocampal circuit by establishing synapses with existing neurons (*Espósito et al., 2005*; *Ge et al., 2006*; *Toni et al., 2008*; *Dieni et al., 2013*). During a transient period of maturation, new GCs exhibit intrinsic and synaptic properties distinct from mature GCs, potentially underlying the contribution of neurogenesis to memory encoding (*Schmidt-Hieber et al., 2004*; *Ge et al., 2007*; *Aimone et al., 2011*; *Marín-Burgin et al., 2012*; *Dieni et al., 2013*; *Brunner et al., 2014*; *Dieni et al., 2016*). Yet computational models also suggest that remodeling of pre-existing circuits by continual neurogenesis can degrade established memories (*Weisz and Argibay, 2012*; *Chambers et al., 2004*), a possibility that has recently gained experimental support from the observation that neurogenesis facilitates 'forgetting' (*Akers et al., 2014*; *Epp et al., 2016*). Circuit remodeling could occur by synaptic redistribution, wherein existing terminals that synapse onto mature GCs are appropriated by newly integrating GCs. This possibility is supported by anatomical evidence that immature dendritic spines transiently receive a high proportion of synapses from multiple-synapse boutons (*Toni et al., 2007*; *Toni and Sultan, 2011*). Furthermore, dramatically increasing the number of new neurons does not alter the density of spines and synapses in the molecular layer, suggesting a readjustment of synaptic connections (*Kim et al., 2009*). Yet whether synaptic integration of new GCs is accompanied by changes in synaptic function and structure of mature GCs is not known.

The number of integrating new GCs can be selectively altered by genetic manipulations targeted to adult stem cells that regulate the survival of progeny (*Enikolopov et al., 2015*). Adult-born neurons undergo a period of massive cell death during the first weeks after cell birth that is rescued by deletion of the pro-apoptotic protein Bax (*Sun et al., 2004*; *Kim et al., 2009*), and conditional *Bax*

**eLife digest** Neurogenesis, the creation of new brain cells called neurons, occurs primarily before birth. However, a region of the brain called the dentate gyrus, which is involved in memory, continues to produce new neurons throughout life. Recent studies suggest that adding neurons to the dentate gyrus helps the brain to distinguish between similar sights, sounds and smells. This in turn makes it easier to encode similar experiences as distinct memories.

The brain's outer layer, called the cortex, processes information from our senses and sends it, along with information about our location in space, to the dentate gyrus. By combining this sensory and spatial information, the dentate gyrus is able to generate a unique memory of an experience. But how does neurogenesis affect this process? As the dentate gyrus accumulates more neurons, the number of neurons in the cortex remains unchanged. Do some cortical neurons transfer their connections – called synapses – to the new neurons? Or does the brain generate additional synapses to accommodate the newborn cells?

Adlaf et al. set out to answer this question by genetically modifying mice to alter the number of new neurons that could form in the dentate gyrus. Increasing the number of newborn neurons reduced the number of synapses between the cortex and the mature neurons in the dentate gyrus. Conversely, killing off newborn neurons had the opposite effect, increasing the strength of the synaptic connections to older cells. This suggests that new synapses are not formed to accommodate new neurons, but rather that there is a redistribution of synapses between old and new neurons in the dentate gyrus.

Further work is required to determine how this redistribution of synapses contributes to how the dentate gyrus works. Does redistributing synapses disrupt existing memories? And how do these findings relate to the effects of exercise – does this natural way of increasing neurogenesis increase the overall number of synapses in the system, potentially creating enough connections for both new and old neurons?

deletion in Nestin- expressing progenitors enhances the number of adult-born neurons without affecting other cell populations (*Sahay et al., 2011*; *Ikrar et al., 2013*). Similarly, inducible expression of the diphtheria toxin receptor in Nestin-expressing stem cells allows selective ablation of adult-born neurons (*Arruda-Carvalho et al., 2011*). These approaches have been used to identify contributions of adult born neurons in hippocampal-based behaviors, with the understanding that behavioral outcomes could either reflect unique functions of adult-born neurons themselves or homeostatic adaptions within the network (*Singer et al., 2011*). Physiological stimuli like exercise and environmental enrichment also enhance dentate neurogenesis, yet it is unclear whether genetically targeted manipulations of neurogenesis mimic the circuit function in the same manner as physiological stimuli.

To identify network adaptions resulting from synaptic integration of new GCs, here we tested how manipulating the number of adult-born GCs affects perforant path-evoked excitatory synaptic currents (EPSCs) in mature GCs. We measured synaptic transmission to pre-existing mature GCs in response to selective genetic manipulations of Nestin-expressing stem cells, using inducible *Bax* deletion to enhance, or diphtheria toxin-induced ablation to reduce, the number of new neurons. We also tested synaptic transmission to immature GCs and mature GCs with *Bax* deletion to investigate potential non-apoptotic functions of the Bax signaling pathway in synaptic function (*Jiao and Li, 2011*; *Ertürk et al., 2014*). Finally, we tested whether enhancing neurogenesis by a physiological stimulus likewise alters excitatory transmission to mature neurons. Our results show that selectively manipulating the number of immature GCs modifies synaptic function of mature GCs in a manner consistent with synaptic redistribution, with an inverse relationship between the number of new neurons and perforant-path evoked EPSCs. In contrast, enhancing neurogenesis via the non-selective paradigm of environmental enrichment generates a net increase in functional connectivity of mature neurons. Together these results demonstrate the capacity of mature GCs to alter synaptic function in response to genetic and experiential circuit manipulations.

## Results

### Enhancing immature neurons decreases EPSCs and spine density of mature neurons

We sought to test synaptic transmission to mature GCs after selectively enhancing the number of integrating new GCs by manipulating cell survival, given that most proliferating DG progenitors and newborn neurons undergo apoptosis (*Sierra et al., 2010*). Cell death of progenitors and new GCs requires the pro-apoptotic protein Bax, a member of the BCL-2 family of proteins in the intrinsic apoptotic pathway (*Sun et al., 2004*). Both germ line and conditional *Bax* deletion block cell death of adult-generated GCs without altering proliferation or the gross structural integrity of the DG (*Sun et al., 2004*; *Kim et al., 2009*; *Sahay et al., 2011*). As previously described (*Sahay et al., 2011*; *Ikrar et al., 2013*), we increased the population of adult-born GCs by crossing inducible Nestin-CreER$^{T2}$ mice with a *Bax* conditional knockout mouse line to selectively block apoptotic cell death in proliferating cells and their progeny (Materials and methods; *Figure 1—figure supplement 1A*). Four-to-six weeks after tamoxifen-induced recombination at two months of age, we compared the number of new GCs and synaptic responses from pre-existing mature GCs in hippocampal slices from $BaxKO_{immature}$ mice (referred to as $BaxKO_{im}$) and controls (*Figure 1A*). We crossed some BaxKO$_{im}$ mice with a transgenic reporter line that labels early postmitotic GCs (*Overstreet et al., 2004*) to reveal a ~40% increase in the number of newborn GCs and overtly normal dentate structure (*Figure 1B,C*).

To assess excitatory transmission from entorhinal cortex across the population of GCs and onto individual mature GCs, we stimulated the medial perforant path while simultaneously recording field excitatory postsynaptic potentials (fEPSPs) and excitatory postsynaptic currents (EPSCs) from mature GCs (*Figure 1D,E*). All experiments were performed in the GABA$_A$ receptor antagonist picrotoxin to isolate glutamatergic synaptic responses. There was no difference in fiber volleys (FVs; a measure of axonal activation) or fEPSPs between slices from BaxKO$_{im}$ and control mice (*Figure 1—figure supplement 2A*) (*Sahay et al., 2011*), as well as no difference in fEPSPs when responses were binned by the FV to account for differences in the number of stimulated axons across slices (*Figure 1F*). We targeted mature GCs located near the mid or outer edge of the granule cell layer and confirmed their maturity by morphology and intrinsic membrane properties (*Figure 1—figure supplement 2B,C*). Interestingly, we found that mature GCs in BaxKO$_{im}$ mice exhibited smaller EPSCs than mature GCs in controls across all FV amplitudes (*Figure 1G*, left), and an overall lower EPSC/FV ratio (*Figure 1G*, right). There was no difference in the EPSC/FV ratio between mature GCs in Cre$^+$ and Cre$^-$ controls, and the difference in EPSCs persisted when only $Bax^{fl/fl}$ genotypes were analyzed (*Figure 1—figure supplement 1B,C*). Thus mature GCs in BaxKO$_{im}$ slices had reduced excitatory transmission.

To assess the pre- or postsynaptic locus of reduced EPSCs in mature GCs from BaxKO$_{im}$ mice, we first compared the paired-pulse ratio (PPR), a measure of presynaptic release probability. There was no difference in the PPR of evoked EPSCs at an interstimulus interval of 100 ms (*Figure 2A*), implying that adult-born neurons do not regulate transmission to mature GCs by secreting a factor that alters the release probability. However, mature GCs in BaxKO$_{im}$ mice displayed a lower frequency of spontaneous EPSCs (sEPSCs) with no change in amplitude (*Figure 2B*), suggesting a reduction in the number of active synapses with no change in postsynaptic responsiveness. Furthermore, using Sr$^{2+}$ to desynchronize evoked release in order to detect single site EPSCs (*Bekkers and Clements, 1999*; *Rudolph et al., 2011*; *Williams et al., 2015*), we found a reduction in the frequency but not the amplitude of desynchronized events (*Figure 2C*). Thus, enhanced numbers of newly generated neurons were associated with reduced excitatory synaptic transmission to mature GCs that appeared to be mediated by fewer functional synapses.

To further examine the locus of change, we assessed the PPR of EPSCs in mature GCs across a range of interstimulus intervals (20–1000 ms). In this protocol, mature GCs in BaxKO$_{im}$ and control mice again exhibited similar passive and active properties (*Figure 3—figure supplement 1*). The PPR was mildly depressing (*Petersen et al., 2013*), with no difference in ratios between genotypes (*Figure 3A*), as previously reported using fEPSPs (*Sahay et al., 2011*). During the recordings, we filled GCs with biocytin for posthoc spine analysis, focusing on dendrite segments in the middle molecular layer where medial perforant path synapses are located (*Figure 3B*). Consistent with reduced evoked and sEPSCs, there was a robust reduction in the density of spines in mature GCs from BaxKO$_{im}$ mice compared to controls (*Figure 3C*). We classified spines by shape (mushroom,

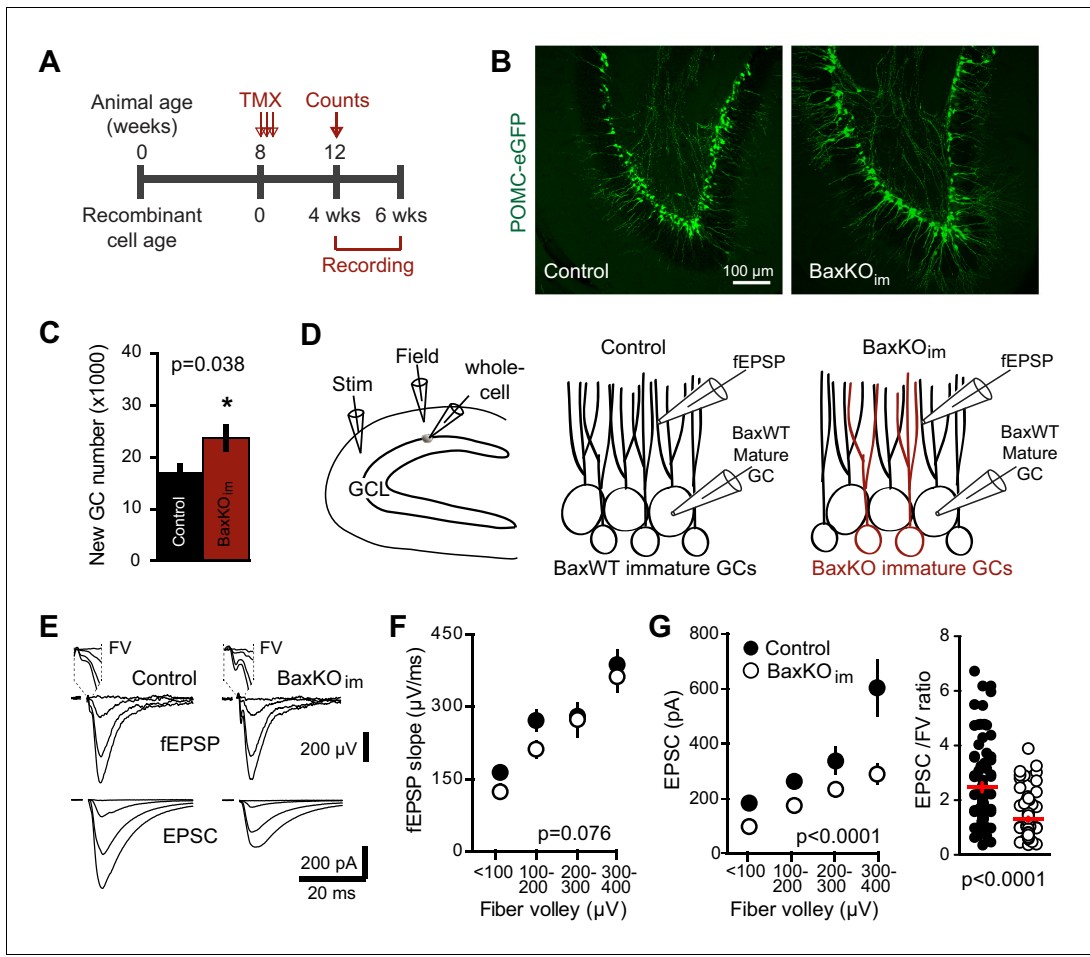

**Figure 1.** Increasing neurogenesis reduces EPSCs in mature GCs. (**A**) The experimental timeline showing recording 4–6 weeks after tamoxifen (TMX)-induced *Bax* deletion in Nestin-expressing progenitors. (**B**) Confocal images of newborn neurons expressing eGFP in fixed sections (50 μm) from control and BaxKO$_{im}$ mice. (**C**) Stereological cell counts of eGFP+ newborn cells revealed neurogenesis was enhanced by 41% (control 16,881 ± 1422 cells, n = 4; BaxKO$_{im}$ 23,756 ± 2166 cells, n = 4; unpaired t-test p=0.038). (**D**) Schematic showing experimental paradigm, with simultaneous fEPSPs and whole-cell recordings of EPSCs from mature GCs. All experiments were performed in the presence of picrotoxin to block GABA$_A$ receptor-mediated currents. (**E**) Examples of fEPSPs (top) with the fiber volley (FV, top inserts) and EPSCs (bottom) in slices from control and BaxKO$_{im}$ mice. Synaptic responses were evoked by increasing intensity stimulation by a patch pipette placed in the middle molecular layer. fEPSPs and EPSCs were binned by FV amplitude. Stimulus artifacts are blanked for clarity. (**F**) The fEPSP versus FV plot illustrates the effectiveness of FV normalization, with fEPSP increasing linearly with axonal recruitment. There was no difference in fEPSPs in slices from BaxKO$_{im}$ and control mice (two-way ANOVA, 0.076). FVs are binned by 100 μV and each symbol denotes the mean and SEM of 10–38 responses from 15 control and 14 BaxKO$_{im}$ slices (with four responses in the largest 300–400 μV FV control bin). (**G**) Left, a decrease in synaptic strength to mature GCs was revealed by the EPSC plotted against FV amplitude (two-way ANOVA, $F_{genotype\ (1,167)}$=54.41 p<0.0001; p<0.05 for all bins with Bonferroni post-tests). Right, the overall EPSC/FV ratio was reduced in BaxKO$_{im}$ slices (unpaired t-test, n = 86, 95).

The following figure supplements are available for figure 1:

**Figure supplement 1.** Generation of BaxKO$_{immature}$ mice.

**Figure supplement 2.** No change in fEPSPs or mature GC intrinsic excitability in BaxKO$_{im}$ mice.

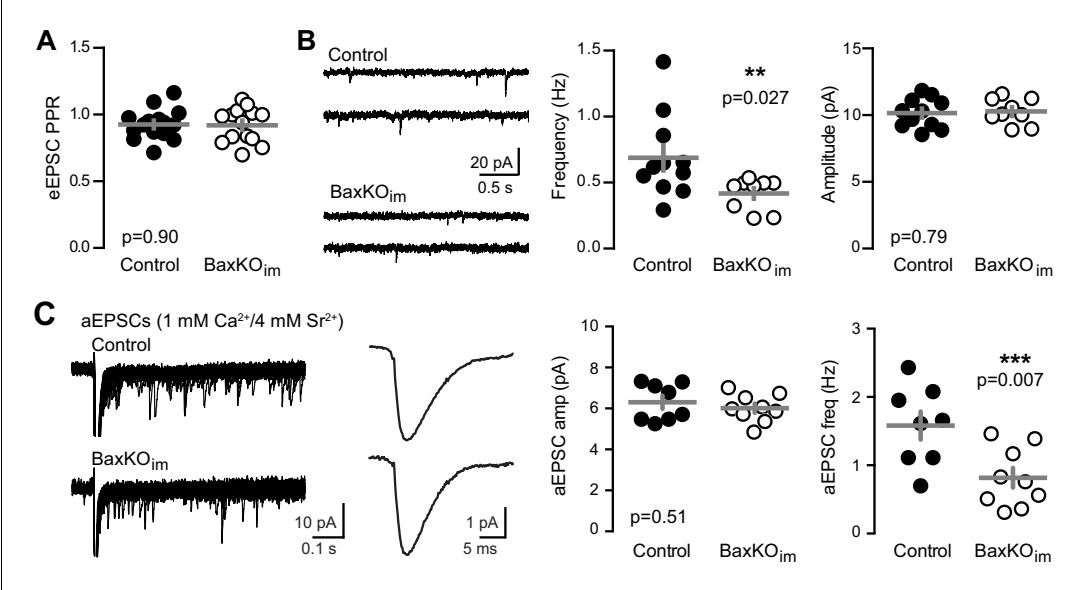

**Figure 2.** Fewer functional synapses on mature GCs in BaxKO$_{im}$ mice. (**A**) The paired-pulse ratio of evoked EPSCs (100 ms ISI) was similar in BaxKO$_{im}$ and control mature GCs (unpaired t-test p=0.90; n = 15 controls,14 BaxKO$_{im}$). (**B**) Spontaneous EPSCs in mature GCs from BaxKO$_{im}$ mice had lower frequency and similar amplitudes as sEPSCs in mature GCs from control mice (unpaired t-test p=0.027 and 0.79, respectively; n = 11 controls, 9 BaxKO$_{im}$). (**C**) Asynchronous EPSCs were generated by desynchronizing synaptic release with 1 mM Ca$^{2+}$ and 4 mM Sr$^{2+}$. Uniquantal aEPSCs were detected following the synchronous EPSC. Left, 40 traces overlaid with examples of averaged aEPSCs. Middle, there was no difference in the average amplitude across genotypes but a reduction in the frequency of aEPSCs (1743 events in 8 GCs from controls, 1015 events in 9 GCs from BaxKO$_{im}$; unpaired t-test p=0.51 and 0.007 respectively). There was no difference in the average rise time or decay of aEPSCs (not shown).

thin, stubby) to determine the percentage of each spine type in control and BaxKO$_{im}$ mice. There was a slight increase in the percentage of stubby spines in BaxKO$_{im}$ mice (**Figure 3D**), with no significant difference in the percentage of thin and mushroom spines. Together, these results support the functional data showing that increasing the number of newborn GCs decreases synaptic transmission to mature GCs by reducing the number of synapses.

## Ablation of immature neurons increases synaptic transmission to mature neurons

We next tested whether genetically ablating adult-generated neurons alters excitatory transmission to mature GCs. We crossed Nestin-CreER$^{tm4}$ mice (**Kuo et al., 2006a**) to Cre-inducible diphtheria toxin receptor (iDTR) mice (**Buch et al., 2005**; **Arruda-Carvalho et al., 2011**). Six weeks after tamoxifen-induced recombination, DT injections were given to ablate immature adult-born GCs in *Nestin-Cre$^+$/iDTR$^+$* offspring (termed Ablated$_{im}$ mice; **Figure 4A**) with *Cre$^-$* littermates used as controls. Ten days after injections, there was a 27% reduction in the number of Dcx-expressing immature cells in the dentate of Ablated$_{im}$ mice (**Figure 4B**; 5601 ± 262, n = 2, compared to 7648 ± 332, n = 4, p=0.016), noting that re-population of Dcx-expressing cells in the period after DT injection can lead to an underestimation of ablation efficiency (**Vukovic et al., 2013**; **Yun et al., 2016**). Performing simultaneous field and whole-cell recordings from mature GCs in Ablated$_{im}$ mice and controls at 1–2 weeks after DT injections suggested no change in total synapses, assayed by the FV and fEPSP slopes (**Figure 4C,D**, **Figure 4—figure supplement 1A**). We also assayed synaptic terminals by immunodetection of the vesicular glutamate transporter (vGlut1) in the molecular layer, and found no differences between controls and either Ablated$_{im}$ or BaxKO$_{im}$ slices (**Figure 4—figure supplement 1B**). Furthermore, there was no change in the fEPSP normalized to the FV (**Figure 4D**). However, there was enhanced synaptic transmission to individual mature GCs, shown by larger EPSC amplitudes across FVs (**Figure 4E**) and an overall larger EPSC/FV ratio (2.2 ± 0.1 in control compared to 3.7 ± 0.4 in Ablated$_{im}$ mice; n = 42, 47 respectively, p=0.001 unpaired t-test). The change in synaptic strength was not associated with any changes in the intrinsic properties of mature GCs

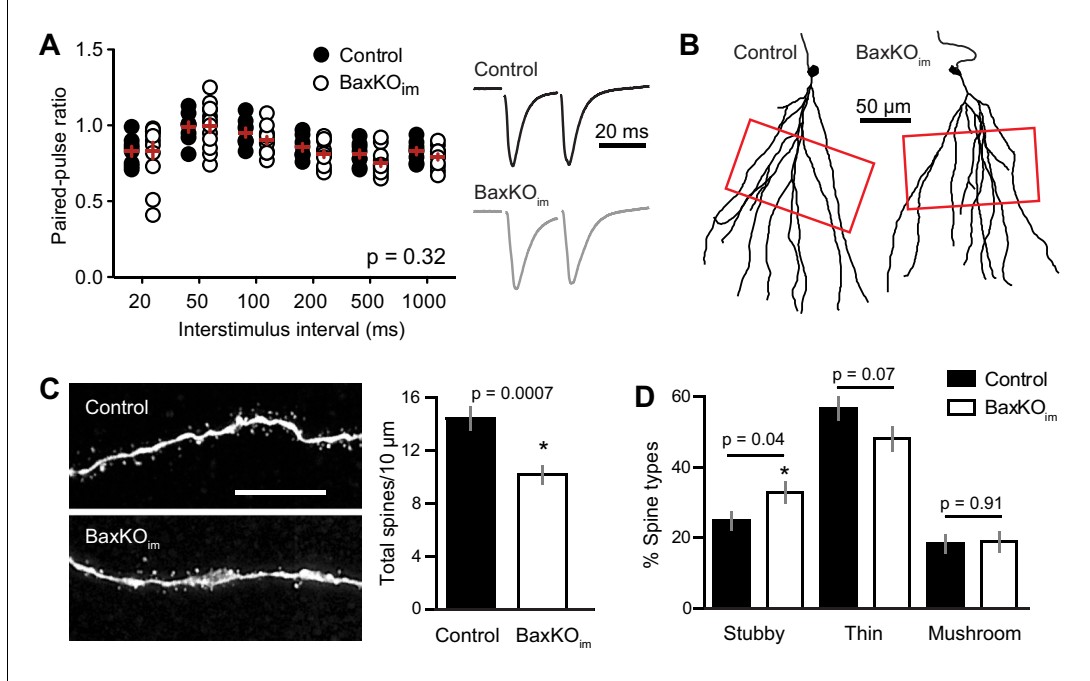

**Figure 3.** Mature GCs in BaxKO$_{im}$ mice exhibit low spine density. (A) There was no difference in the paired-pulse ratio of EPSCs in mature GCs from BaxKO$_{im}$ and control mice across a range of interstimulus intervals (2-way ANOVA, p=0.31, n = 8,12 mature GCs). (B) Examples of reconstructed mature GCs from the recordings in (A). Red boxes indicate regions used for spine analysis. (C) Left, example images of dendritic spines from mature GCs. Scale bar, 10 µm. Middle, the density of dendritic spines was lower in BaxKO$_{im}$ mice (14 ± 0.8 spines/10 µm, 936 total spines counted on 15 dendritic segments in two control mice; 10 ± 0.6 spines/10 µm, 676 total spines on 12 dendritic segments from 3 BaxKO$_{im}$ mice; p=0.0007 unpaired t-test). (D) Classifying spines as stubby, thin and mushroom revealed a significant increase in the percentage of stubby spines in mature GCs from BaxKO$_{im}$ mice (p=0.04 unpaired t-test) with no change in the percentage of thin spines (p=0.07 unpaired t-test) or mushroom spines (p=0.45 unpaired t-test).

The following figure supplement is available for figure 3:

**Figure supplement 1.** Intrinsic properties of mature GCs for PPR and spine analysis.

(*Figure 4—figure supplement 2A*). These results suggest that reducing the number of immature GCs increases the strength of synaptic transmission to mature GCs, an effect that cannot be explained by altered inhibition as GABA$_A$ receptors were blocked in these experiments (*Singer et al., 2011*; *Temprana et al., 2015*; *Drew et al., 2016*). There was no difference in PPR, suggesting that release probability was unchanged (*Figure 4—figure supplement 2B*). We were unable to detect differences in the average frequency or amplitude of sEPSCs in mature GCs from Ablated$_{im}$ mice (*Figure 4—figure supplement 2C*), making it unclear whether reduced EPSCs resulted from pre- or postsynaptic mechanisms. Since the frequency of spontaneous activity in GCs is low, the threshold for detecting differences in synaptic function using spontaneous activity may be higher than for evoked transmission with FV normalization, and it appears that neurogenesis was altered by a greater degree in BaxKO$_{im}$ mice compared to Ablated$_{im}$ mice (~40% versus 25% change in new neuron number). However, we also cannot rule out the possibility that separate pools of synaptic vesicles contribute to differences between results obtained with evoked and spontaneous assays (reviewed in *Kavalali, 2015*).

In summary, manipulating the number of immature GCs was inversely associated with excitatory synaptic strength of mature GCs. These manipulations did not affect global measures of axonal activation, synaptic strength or presynaptic terminals, suggesting that changing the number of newly generated neurons did not alter the total number of afferent axons or synapses. The idea that global measures of basal synaptic transmission and release probability are independent of the number of dentate GCs is in agreement with prior results in the conditional *Bax*KO (*Sahay et al., 2011*) as well as the observation that perforant path synapse density is unaltered in germline *Bax*KO mice

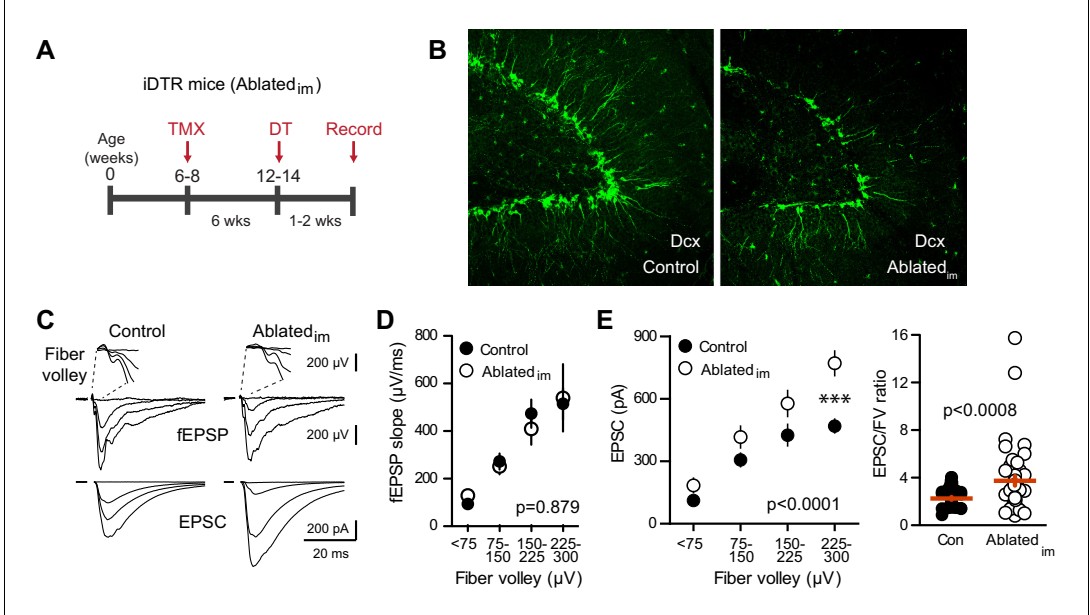

**Figure 4.** Ablating neurogenesis increases synaptic transmission to mature GCs. (A) Experimental timeline showing ablation of immature GCs that are <6 weeks of age. Recordings from mature GCs were done 1–2 weeks after ablation. (B) Confocal images of Dcx-expressing immature neurons in control and Ablated$_{im}$ mice. (C) Example of fEPSPs (top) with fiber volleys (FV, top insets) and simultaneously recorded EPSCs from mature GCs (bottom) in control and Ablated$_{im}$ mice. (D) There was no difference in the fEPSP slope versus FV between Ablated$_{im}$ and control mice (two-way ANOVA p=0.879, each symbol represents 8–22 responses from 7 control and 7 Ablated$_{im}$ mice; FVs were binned by 75 μV). (E) The EPSC amplitude plotted against FV was larger in mature GCs from Ablated$_{im}$ mice compared to controls (two-way ANOVA, F$_{genotype (1,91)}$=30.31 p<0.0001; ***p<0.001 Bonferonni post-test). There was an increase in the overall EPSC/FV ratio in mature GCs from Ablated$_{im}$ mice (unpaired t-test, p=0.0008, n = 42, 47).

The following figure supplements are available for figure 4:

**Figure supplement 1.** No change in FV, fEPSP slope or vGlut1 expression.

**Figure supplement 2.** Unlabeled GCs in Ablated$_{im}$ mice have mature intrinsic properties and no change in PPR or sEPSCs.

which exhibit dramatically enhanced numbers of dentate GCs (*Kim et al., 2009*). Together these results support the idea that synaptic integration of newborn GCs involves a redistribution of existing synapses from old to new cells (*Tashiro et al., 2006*; *Toni et al., 2007*; *McAvoy et al., 2016*).

### *Bax* deletion enhances synaptic strength of immature neurons

One assumption inherent to this idea, however, is that synaptic integration of newborn neurons is unaffected by manipulating their number, such that the increase in new cell number is paralleled by an increase in the total number of new synapses. We therefore sought to confirm synaptic integration of *Bax*KO immature GCs by crossing *Bax*KO$_{im}$ and control mice with a tdTomato reporter line (Ai14) to target *Bax*KO and *Bax*WT immature GCs for recordings (*Figure 5A*). The input resistance is a measure of cell maturity (*Overstreet-Wadiche and Westbrook, 2006*; *Dieni et al., 2013*) and as expected, labeled immature GCs (six weeks post-tamoxifen) had higher input resistance than mature GCs, with no difference between genotypes (*Figure 5B*). This confirms that the immature GCs were at a similar stage of maturation and is consistent with the similar dendrite development reported in this model (*Sahay et al., 2011*). FVs and fEPSP slopes were the same between genotypes, replicating the results of *Figure 1* and further suggesting a similar level of axonal activation and number of total synapses after conditional *Bax* deletion (*Figure 5—figure supplement 1*). Consistent with the low excitatory connectivity of immature GCs (*Dieni et al., 2016*), in control mice the EPSC/FV ratio of immature GCs (1.24 ± 0.07 n = 80) was lower than the EPSC/FV ratio in mature GCs (2.44 ± 0.16 n = 86, p<0.0001 unpaired t-test). But unexpectedly, simultaneously recorded fEPSPs and EPSCs revealed that EPSCs in *Bax*KO immature GCs were significantly larger than EPSCs in *Bax*WT

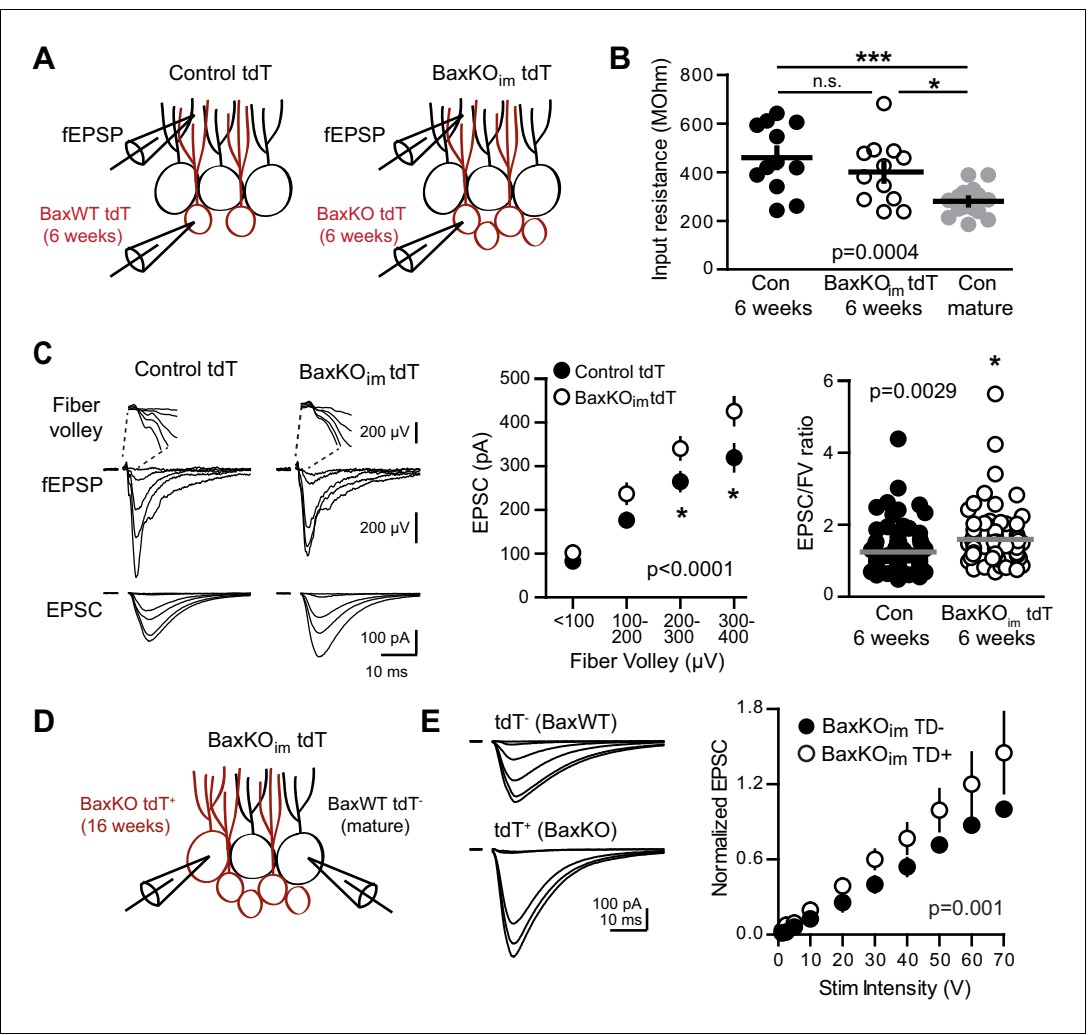

**Figure 5.** *Bax* deletion enhances EPSCs in adult born neurons. (**A**) Whole cell recordings were made from immature GCs in control and BaxKO$_{im}$ slices at six weeks post-tamoxifen injection, using picrotoxin to isolate glutamatergic EPSCs. Simultaneous fEPSPs were recorded in the molecular layer as in *Figure 1*. (**B**) Immature GCs in control and BaxKO$_{im}$ tdT mice had a similar input resistance that was higher than mature GCs (n = 12, 12, 16, respectively; one-way ANOVA p=0.0004, *p<0.05,***p<0.0001 Bonferroni post hoc test). (**C**) Left, examples of fEPSPs (top) and EPSCs (bottom) recorded in immature GCs. Middle, an increase in synaptic transmission to immature *Bax*KO GCs was revealed by the EPSC plotted against fiber volley (two-way ANOVA, $F_{genotype\ (1,143)}$=18.55 p<0.0001, n = 12 control tdT, 12 BaxKO$_{im}$ tdT; *p<0.05 with Bonferroni post-tests). Right, the EPSC/FV ratio for all stimulus intensities (control 1.24 ± 0.07, n = 80; BaxKO$_{im}$1.59 ± 0.09, n = 75; unpaired t-test p=0.0029). (**D**) Schematic showing simultaneous recordings from adjacent tdT⁻ (*Bax*WT) and tdT⁺. (*Bax*KO) GCs in slices from BaxKO$_{im}$ tdT mice at 16 weeks after tamoxifen. (**E**) Adult-generated *Bax*KO GCs had larger EPSCs than simultaneously recorded unlabeled mature GCs. EPSCs were normalized to the maximum amplitude of the unlabeled (*Bax*WT) GC in each slice (two-way ANOVA, $F_{genotype\ (1,94)}$=11.59 p=0.001, n = 6 pairs), scale bars: 10 ms, 100 pA. Comparing raw EPSCs between pairs of unlabeled and tdT⁺ GCs across all stimulus intensities confirmed EPSCs were larger in tdT⁺ GCs (not shown, paired t-test, p<0.0013).

The following figure supplements are available for figure 5:

**Figure supplement 1.** No change in FV or fEPSP in slices from BaxKO$_{im}$ tdT mice.

**Figure supplement 2.** No change in PPR or sEPSCs in immature GCs from BaxKO$_{im}$ tdT mice.

**Figure supplement 3.** No differences in intrinsic properties between adult-born mature tdT⁺ (*Bax*$^{-/-}$) and unlabeled mature GCs.

*Figure 5 continued on next page*

*Figure 5 continued*

**Figure supplement 4.** Global Bax levels are unaltered in BaxKO$_{im}$ hippocampus.

immature GCs across FV bins, and the overall EPSC/FV ratio was greater (*Figure 5C*). Thus *Bax*KO immature GCs showed enhanced synaptic transmission compared to WT immature GCs. The PPR of EPSCs in immature GCs was similar between genotypes and there was not a significant difference in the frequency or amplitude of sEPSCs (*Figure 5—figure supplement 2*), again noting that the low frequency of spontaneous activity in immature GCs (*Mongiat et al., 2009*; *Dieni et al., 2016*) makes it difficult to interpret the lack of change in sEPSCs. These results confirm that *Bax*KO immature GCs acquired synapses during integration and, in fact, suggest *Bax* deletion promotes the synaptic integration of new GCs.

To further test the role of Bax in excitatory transmission to postmitotic GCs, we compared synaptic activity of adult-born *Bax*KO and unlabeled GCs at 16 weeks after tamoxifen-induced recombination, well after excitatory synaptic integration is complete (*Mongiat et al., 2009*). We directly compared EPSCs using simultaneous recordings from neighboring *Bax*WT (tdT$^-$) and *Bax*KO (tdT$^+$) GCs; *Figure 5D*). In this paradigm, FV normalization is unnecessary because the number of stimulated axons is the same for both recorded cells. To compare across cell pairs with different numbers of stimulated fibers in each slice, we normalized EPSCs to each *Bax*WT GC. Consistent with a role of Bax suppressing synaptic depression, EPSCs in *Bax*KO GCs were larger than EPSCs in *Bax*WT GCs (*Figure 5E*). There was no difference in the mature intrinsic properties of *Bax*WT and *Bax*KO GCs, again showing that *Bax* deletion does not alter intrinsic cell properties (*Figure 5—figure supplement 3*). Thus, enhanced synaptic transmission in *Bax* deficient GCs persists when adult-born neurons are fully mature.

## *Bax* deletion in mature neurons increases EPSCs and spine density

Our results show that *Bax* deletion increases excitatory synaptic integration of adult born GCs, consistent with growing evidence that the Bax/caspase signaling cascade has non-apoptotic functions in synaptic plasticity (*Unsain and Barker, 2015*). Prior work suggests that Bax activation is an intermediary step between NMDAR-Ca$^{2+}$ influx and local activation of caspase-3, which in turn is necessary and sufficient for LTD and subsequent spine pruning (*Li et al., 2010*; *Jiao and Li, 2011*; *Ertürk et al., 2014*; *Sheng and Ertürk, 2014*). The high level of *Bax* mRNA throughout the adult dentate gyrus (*Lein et al., 2007*) raises the possibility that this pathway contributes to activity-dependent synaptic remodeling of mature GCs in addition to controlling the number of integrating new GCs via apoptosis. Given that synaptic strength may depend on Bax expression, we tested whether overall Bax levels are altered in BaxKO$_{im}$ mice. Western blot analysis revealed no difference in Bax protein levels in hippocampal lysates from BaxKO$_{im}$ and control mice, showing that deletion of *Bax* from a small percentage of GCs does not lead to widespread changes in Bax protein (*Figure 5—figure supplement 4*).

To further probe the synaptic function of Bax, we next tested whether enhanced synaptic strength persists in mature neurons when *Bax* is deleted from postmitotic GCs throughout development. We generated conditional *Bax*KO in postmitotic GCs (termed BaxKO$_{mature}$) using *POMC-Cre* to direct recombination in dentate GCs throughout development (*Gao et al., 2007*; *Figure 6—figure supplement 1*). Expression of tdTomato (tdT)reporter revealed that most, but not all, NeuN-expressing GCs in the granule cell layer expressed Cre and that NeuN-lacking proliferating progenitors in the subgranular zone were Cre negative (*Figure 6A*), consistent with transient activity of the *POMC* promoter in early postmitotic GCs (*Overstreet-Wadiche et al., 2006*; *Overstreet et al., 2004*). We compared EPSCs in simultaneous recordings from neighboring tdT$^+$ (*Bax*KO) and tdT$^-$ (*Bax*WT) mature GCs (*Figure 6B*), again normalizing EPSCs to each WT cell to compare EPSCs across cell pairs. EPSCs in *Bax*KO GCs were larger than EPSCs in *Bax*WT GCs across a range of stimulus intensities (*Figure 6C*). To confirm that the increase in EPSC amplitude resulted from *Bax* deletion, we repeated the experiment in *POMC-Cre/Bax*WT/tdT mice (*Figure 6D,E*). EPSCs were the same in neighboring tdT$^+$ and tdT$^-$ mature GCs (*Figure 6F*), indicating that the difference shown in

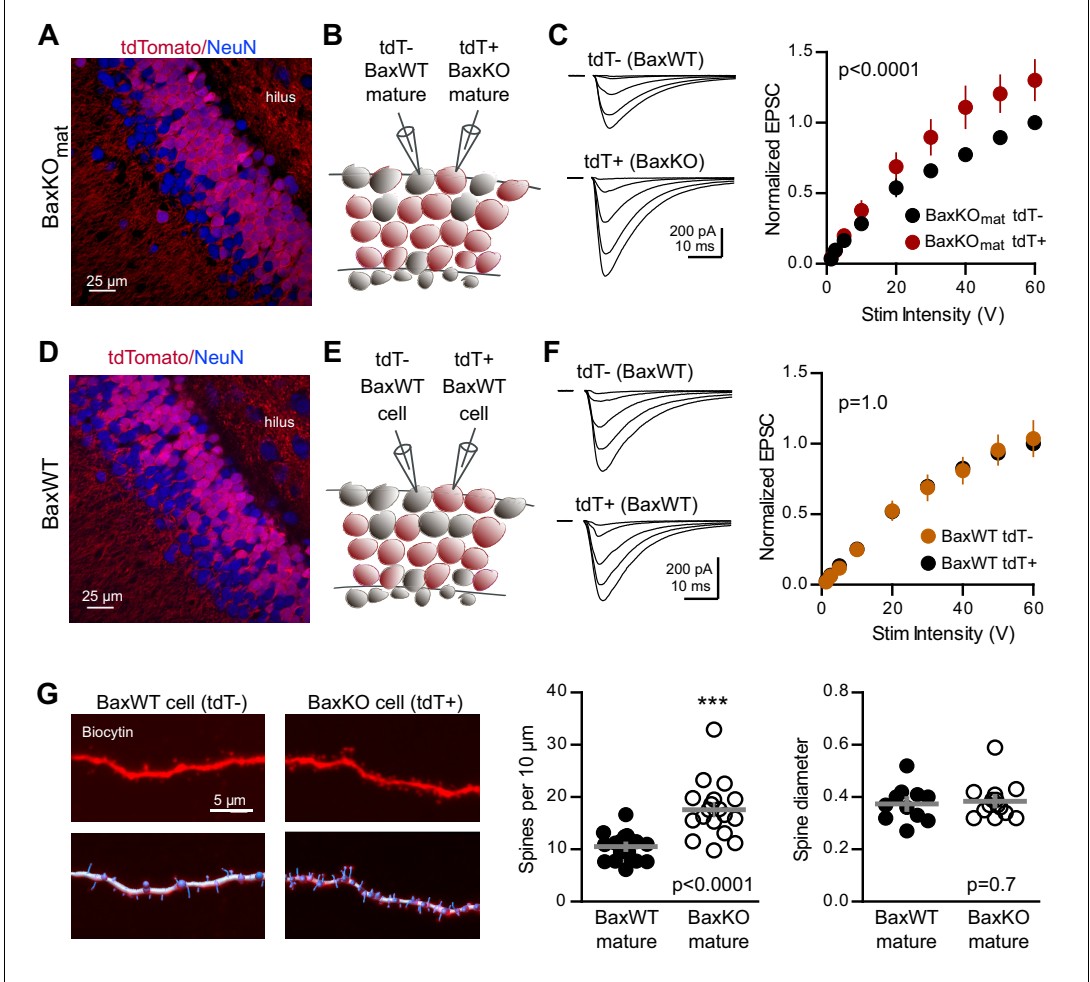

**Figure 6.** *Bax* deletion increases EPSCs and spine density of mature GCs. (**A**) Confocal image of fixed tissue from a BaxKO_mat /tdTomato mouse showing tdT (red) and NeuN (blue). Note the larger fraction of tdT$^+$ GCs compared to (**D**), consistent with enhanced survival of GCs that lack Bax (confirmed in *Figure 7A*). (**B**) Adjacent tdT$^-$ (*Bax*WT) and tdT$^+$ (*Bax*KO) mature GCs were recorded simultaneously. (**C**) Examples of EPSCs in tdT$^-$ and tdT$^+$ mature GCs to the same stimuli. EPSCs were normalized to the maximum EPSC of the unlabeled cell in each slice. EPSCs were larger in tdT$^+$ GCs (two-way ANOVA, F_{genotype (1,198)}=21.14 p<0.0001, n = 12 cell pairs). (**D**) Confocal image of fixed tissue from a *Bax*WT/*POMC-Cre*$^+$/tdTomato mouse, in which both tdT$^+$ and unlabeled GCs are *Bax*WT(red tdT, blue NeuN). (**E**) Adjacent tdT$^+$ and unlabeled mature GCs were recorded simultaneously. (**F**) There was no difference in EPSCs between *Bax*WT tdT$^+$ and unlabeled cells (two-way ANOVA p=1.0, n = 8 cell pairs), confirming the difference in panel C requires the *Bax*$^{-/-}$ genotype. (**G**) Posthoc dendrite reconstructions (top) revealed higher spine density in *Bax*KO GCs from (**A**) (10.5 ± 0.53 spines/10 μm in *Bax*WT compared with 17.60 ± 1.3 *Bax*KO, unpaired t-test p<0.0001) with no change in spine head diameter (unpaired t-test, p=0.7, n = 21 segments from 5 *Bax*WT, 18 segments from 9 *Bax*KO). Lower images illustrate spine analysis.

The following figure supplements are available for figure 6:

**Figure supplement 1.** Generation of BaxKO_mature mice.

**Figure supplement 2.** No difference in Ca$^{2+}$-dependence of evoked EPSCs in BaxKO_mat mice.

*Figure 6C* requires the *Bax*$^{fl/fl}$ genotype. Thus, *Bax* deletion from immature GCs *decreases* EPSCs in mature GCs via a non-cell autonomous mechanism (*Figures 1–3*), whereas here we show a cell-autonomous effect of *Bax* deletion that *increases* EPSCs in mature GCs (*Figure 6*). These counterintuitive results could occur if *Bax* deletion generates presynaptic actions that are most evident when *Bax* is deleted from a large population of GCs. We addressed potential presynaptic alterations in BaxKO_mat mice by testing the Ca$^{2+}$-dependence of synaptic transmission. However, we found no

difference in presynaptic function as assessed by comparing EPSC amplitudes and PPRs across a range of extracellular $Ca^{2+}$ concentrations (*Figure 6—figure supplement 2*).

Since Bax activation is necessary and sufficient to activate caspase-3, which acts as a mediator of activity-dependent hippocampal LTD and synaptic pruning (*Li et al., 2010*; *Jiao and Li, 2011*; *Ertürk et al., 2014*; *Lo et al., 2015*), we wondered whether enhanced synaptic transmission to *Bax*KO GCs resulted from a deficit in synaptic pruning. We analyzed dendritic spines in *Bax*KO and *Bax*WT GCs by filling cells with biocytin during recordings. Posthoc analysis revealed a significant increase in the density of spines in *Bax*KO mature GCs, with no change in head diameter (*Figure 6G*). Together these results show that loss of Bax in GCs generates a persistent enhancement of synaptic transmission consistent with a deficit in synaptic pruning.

## Neurogenesis-induced loss of synaptic strength requires intact Bax signaling

Based on the above results, we predicted that neurogenesis-induced loss of synapses from mature GCs might require intact Bax signaling to allow synaptic pruning. We thus assayed neurogenesis-induced synapse loss from mature GCs in BaxKO$_{mat}$ mice, where most mature GCs lack Bax. First, we confirmed that *Bax*KO in newly postmitotic GCs increases the number of integrating new neurons by assessing neurogenesis using *POMC*-eGFP expression. Consistent with the later period of cell death that occurs in newly postmitotic GCs (*Sierra et al., 2010*), we found that the number of newborn integrating neurons was enhanced to a similar degree as observed in BaxKO$_{im}$ mice (*Figure 7A*). However, neurogenesis-induced suppression of synaptic transmission to mature GCs was absent, since the evoked EPSC was similar to controls across all stimulus intensities and the average EPSC/FV ratio was unchanged (*Figure 7B*). Similar to Ablated$_{im}$ and BaxKO$_{im}$ mice, there was no difference in axonal activation or total synapse number, measured by the FV amplitude and fEPSP slope versus FV, respectively (*Figure 7—figure supplement 1A*). Intrinsic properties of mature GCs were the same in BaxKO$_{mat}$ and control mice, showing that *Bax* deletion does not affect these measures of cellular excitability (*Figure 7—figure supplement 1B*). There was also no difference in the PPR, sEPSC frequency or sEPSC amplitude between mature GCs in control and BaxKO$_{mat}$ mice (*Figure 7—figure supplement 2*). However, there was considerable variability in EPSC/FV ratios and sEPSC frequencies in mature GCs from BaxKO$_{mat}$ mice, potentially indicative of the heterogeneous population of *Bax*WT and *Bax*KO GCs (as in *Figure 6A*) with mixed susceptibility to neurogenesis-induced synapse impairment. Together, these results suggest that neurogenesis-induced loss of synaptic strength to mature GCs requires intact Bax signaling.

## Environmental enrichment increases synaptic strength of mature neurons

Our experiments revealed that selective manipulations of adult-born neurons are sufficient to alter functional synaptic transmission to mature neurons, raising the question of whether enhancing neurogenesis by physiological stimuli likewise affects synaptic function of mature neurons. One long-established strategy to enhance neurogenesis is housing rodents with environmental enrichment (EE) that includes exploration of novel objects, social interactions, and running wheels. EE enhances both the number of newborn GCs and their synaptic integration (*van Praag et al., 1999*; *Tashiro et al., 2007*; *Ambrogini et al., 2010*; *Chancey et al., 2013*; *Bergami et al., 2015*), as well as altering structural plasticity in the dentate and other brain regions (*Green and Greenough, 1986*; *Foster et al., 1996*; *Eadie et al., 2005*; *Foster and Dumas, 2001*; *Stranahan et al., 2007*).

We enhanced neurogenesis by housing WT mice with EE (*Figure 8A*), a treatment reported to generate a 1.5–2-fold increase in the number of integrating new GCs (*van Praag et al., 1999*; *Brown et al., 2003*; *Olson et al., 2006*). We previously found that housing mice with running wheels alone for four weeks increases the number of *POMC*-eGFP labeled GCs to 146% of age-matched controls (*Overstreet et al., 2004*), suggesting that EE enhances neurogenesis to a similar or greater extent as observed in BaxKO$_{im}$ mice (*Figure 1B,C*). To assess the strength of excitatory transmission from entorhinal cortex across the population of GCs and onto individual mature GCs, we again stimulated the medial perforant path while simultaneously recording fEPSPs and EPSCs from mature GCs in GABA$_A$ receptor antagonists (*Figure 8B*). As previously reported (*Green and Greenough, 1986*; *Foster et al., 1996*), the fEPSP slope was enhanced in slices from EE mice with no difference

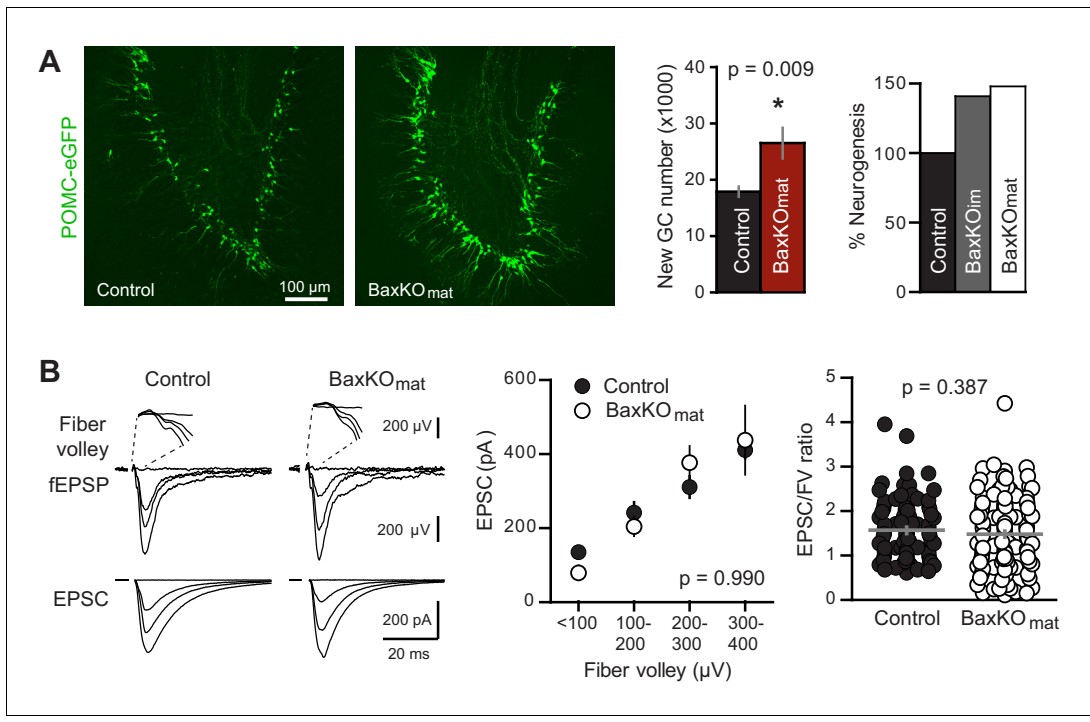

**Figure 7.** Neurogenesis-induced loss of synaptic transmission requires *Bax* in mature GCs. (**A**) Confocal images of newborn neurons expressing eGFP in BaxKO_mat mice. Stereological cell counts revealed neurogenesis was enhanced by ~48% (control 17,910 ± 900 cells, n = 7; BaxKO_mat26,508 ± 2728 cells, n = 6; unpaired t-test), similar to enhanced neurogenesis in BaxKO_im mice. (**B**) Left, examples of fEPSPs (top) and EPSCs in mature GCs (bottom) from control and BaxKO_mat mice. Middle, there was no difference in EPSCs across FVs (two-way ANOVA p=0.990, n = 12 control, 19 BaxKO_mat GCs) or in the EPSC/FV ratio (unpaired t-test p=0.387, n = 88 control, 129 BaxKO_mat), although there was greater variability in the BaxKO_mat group (CV = 52% vs. 43%) consistent with a mixed population of mature $Bax^{-/-}$ and $Bax^{+/+}$ GCs (as shown in **Figure 6**).

The following figure supplements are available for figure 7:

**Figure supplement 1.** No differences in FVs, fEPSP slopes and intrinsic properties of mature GCs in BaxKO_mat mice.

**Figure supplement 2.** Similar PPR and sEPSCs in mature GCs from BaxKO_mat mice.

in the FV, suggesting an increase in total synaptic strength with no change axonal excitability (**Figure 8—figure supplement 1A,B**). Indeed, normalizing the fEPSP slope to the FV to account for differences in the number of stimulated axons across slices revealed a significant increase in the fEPSP (**Figure 8C**). We targeted mature GCs located near the outer edge of the granule cell layer and confirmed their maturity by intrinsic membrane properties (**Figure 8—figure supplement 1C**). Consistent with the enhanced fEPSPs, EPSCs in mature GCs were larger in slices from mice housed in EE (**Figure 8D**), such that the overall EPSC/FV ratio was 2.6 ± 0.16 in EE compared to 1.6 ± 0.07 in control (n = 88, 58 respectively, p<0.0001 unpaired t-test). Enhanced synaptic strength after EE could result either from increased release probability, increased number of synapses or increase in the number of receptors per synapse. We found no difference in the PPR of evoked EPSCs, suggesting that release probability is unchanged (**Figure 8E**), as previously reported (**Foster et al., 1996**). However, the frequency of sEPSCs was increased with no change in sEPSC amplitude (**Figure 8F**), similar to the recently reported increase in miniature EPSCs in mature GCs after EE (**Kajimoto et al., 2016**). Together these results suggest that enhanced evoked EPSCs in mature GCs result from greater number of functional synapses, consistent with increased spine density in Golgi-stained (presumably mature) dentate GCs (**Eadie et al., 2005**; **Stranahan et al., 2007**). These results show that mature GCs exhibit experience-dependent synaptic enhancement that argues against the recently described

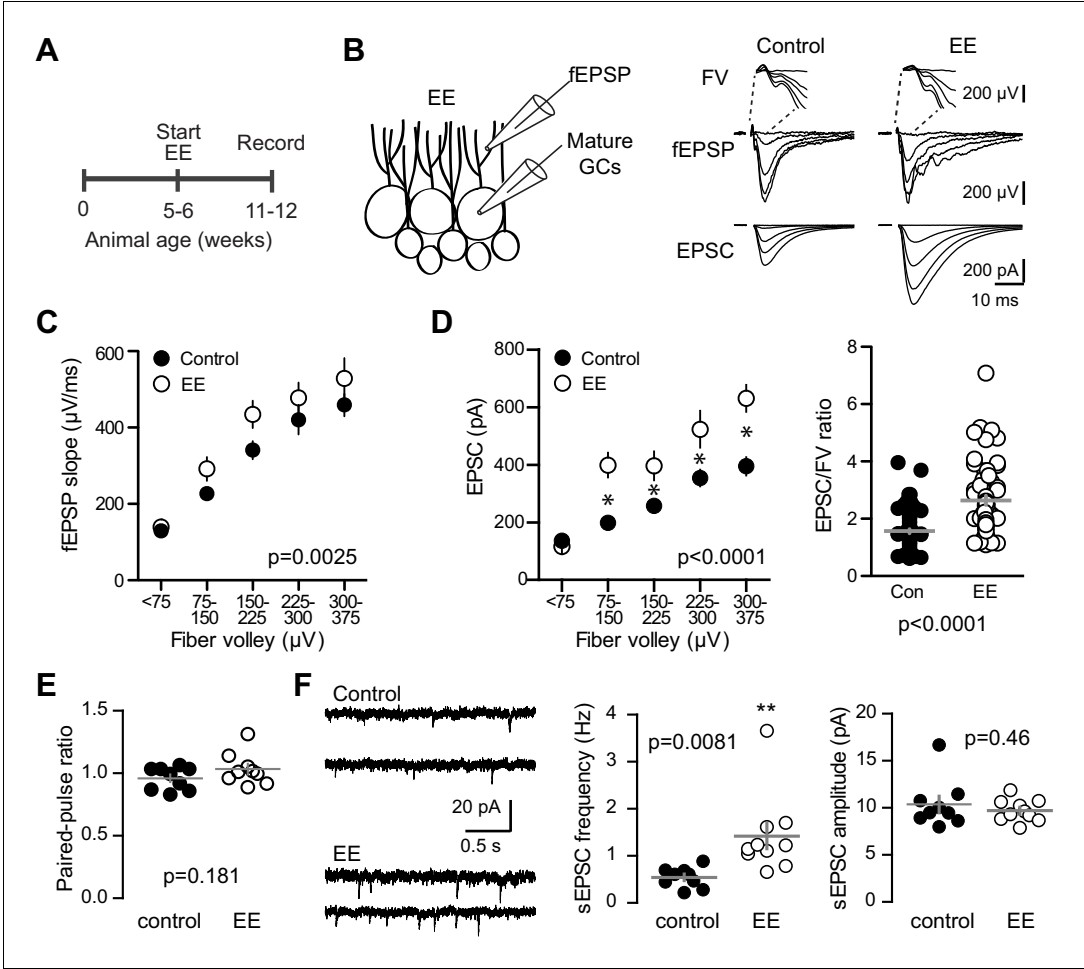

**Figure 8.** Environmental enrichment increases synaptic transmission to mature GCs. (A) The experimental timeline showing recordings performed 4–6 weeks after EE. (B) Left, simultaneous fEPSPs and whole-cell recordings from mature GCs, as shown in *Figure 1*. Examples of fEPSPs (top) with FV (insert) and EPSCs in mature GCs (bottom) in slices from control and EE mice. (C) Slices from EE mice exhibited an increase in the fEPSP slope plotted against FV amplitude (two-way ANOVA, $F_{\text{manipulation} (1,116)}$=9.59, p=0.0025, n = 11 control, 9 EE). FVs were binned by 75 µV. (D) Left, an increase in synaptic transmission to mature GCs was revealed by the EPSC plotted against FV (two-way ANOVA, $F_{\text{manipulation} (1,150)}$=52.88, p<0.0001, n = 11 control, 9 EE). *p<0.01 with Bonferroni post-tests. Right, the overall EPSC/FV ratio was enhanced by EE (unpaired t-test, p<0.0001). (E) The paired-pulse ratio of EPSCs (100 ms ISI) was similar in EE and control mice (p=0.181 unpaired t-test, n = 9 controls, 9 EE). (F) Spontaneous EPSCs in mature GCs from EE mice had higher frequency (p=0.0081 unpaired t-test) but similar amplitude as sEPSCs in mature GCs from control mice (p=0.46, n = 9 controls,10 EE).

The following figure supplement is available for figure 8:

**Figure supplement 1.** EE enhances the fEPSP with no change in fiber volley and input resistance of mature GCs.

restricted period for experience-dependent plasticity of dentate GCs (*Bergami et al., 2015*). However, these results cannot resolve whether integration of EE-induced newborn GCs affects synaptic function of mature neurons. Increased connectivity of mature neurons is likely a parallel phenomenon independent of neurogenesis, since similar increases in synaptic transmission and spine density occur in non-neurogenic regions (*Rampon et al., 2000*; *Malik and Chattarji, 2012*; *Jung and Herms, 2014*). Thus, the magnitude of increased connectivity of mature GCs could be reduced by neurogenesis-induced synaptic redistribution. Altogether, these results highlight the capacity of mature GCs

to undergo changes in synaptic connectivity in response to both genetic and experiential circuit manipulations.

## Quantitative estimate of synapse transfer between mature and immature neurons

Immature GCs make up a small percent of total GCs, and yet when neurogenesis was selectively manipulated the change in synaptic strength to mature GCs was unexpectedly robust. To determine whether the magnitude of altered transmission to mature GCs could be explained by a redistribution of existing synapses to integrating new GCs, we made a quantitative estimate of the proportion of mature synapses that would be transferred to new GCs over the time course of our experiments. We simulated the BaxKO$_{im}$ condition, since in this condition we quantified excitatory input to mature GCs and immature GCs, as well as the increase in new cells induced by *Bax* deletion. Other parameters were based on reported rates of neurogenesis (*Chancey et al., 2013*; *Gil-Mohapel et al., 2013*), cell death (*Sierra et al., 2010*) and excitatory synaptic integration (*Dieni et al., 2013*, *2016*). The simulation is based on a static number of synapses that re-distribute to immature GCs according to their number and time-dependent synaptic integration (*Figure 9—figure supplement 1*). The simulation showed a steep increase in the proportion of synapses occupied by immature GCs in BaxKO$_{im}$ mice starting at the time point when immature GCs start to integrate into the network (*Figure 9A*, red line). The robust transfer of synapses resulted not only from the increased number of immature GCs, but also from the increased acquisition of immature synapses resulting from *Bax* deletion. The predicted reduction in mature synapse number (expressed as a %) at days 36–43 in the simulation was similar to the % change in mature EPSCs measured experimentally (*Figure 9B*). Despite the small proportion of immature GCs within the network (initially set at 5%), the continuous increase in cell number along with enhanced synaptic integration was compounded over time to attenuate synapses on pre-existing neurons to a degree that could account for the magnitude of reduced synaptic strength observed in the BaxKO$_{im}$ experiments.

## Discussion

Here we tested how manipulating the number of adult-born GCs affects excitatory synaptic transmission to mature GCs. We found that selectively manipulating adult-born neurons inversely correlated with synaptic strength of mature neurons with no detectable changes in global measures of synaptic transmission. We reasoned that there are two ways that integrating newborn GCs can acquire synapses; new GCs can form new synaptic connections with existing afferent axons or new GCs can take pre-existing synapses from neighboring mature GCs. If synaptic integration of developing GCs triggers formation of new presynaptic terminals, then neurogenesis will increase the total number of synapses within the DG network but will not affect the number of synapses per mature GC (*Figure 9C1*). In contrast, appropriation of existing synaptic terminals would cause mature GCs to lose synapses while the total number of synapses in the network remains constant (*Figure 9C2*). By comparing measures of total synapses (fEPSPs) and synapses per mature GC (EPSCs) after selectively altering neurogenesis, our results support the latter model wherein newborn GCs appropriate existing synapses and consequently modify synaptic input to mature GCs.

### Enhancing neurogenesis reduces mature neuron synaptic transmission and spine density

Our results showing that increasing neurogenesis decreased synaptic transmission and spine density of mature GCs is consistent with the idea that immature neuron synaptic integration is a competitive process (*Tashiro et al., 2006*; *Toni and Sultan, 2011*; *McAvoy et al., 2016*). Anatomical analysis has suggested that multisynaptic boutons (MSBs) represent an intermediary structure in the transfer of functional synapses from mature to immature GCs (*Toni et al., 2007*; *Toni and Sultan, 2011*). Although we did not find evidence for alterations in the total number of functional synapses reflecting the presence of MSBs when neurogenesis was manipulated, shared transmission from MSBs may be functionally silent due to lack of AMPA receptors on new neurons (*Wu et al., 1996*; *Chancey et al., 2013*), or may be below the detection limits of field potential recordings. Furthermore, recent work suggests MSBs are a common feature of mature GCs and the complexity of MSB innervation increases with GC maturation (*Bosch et al., 2015*), so it is unclear how our functional

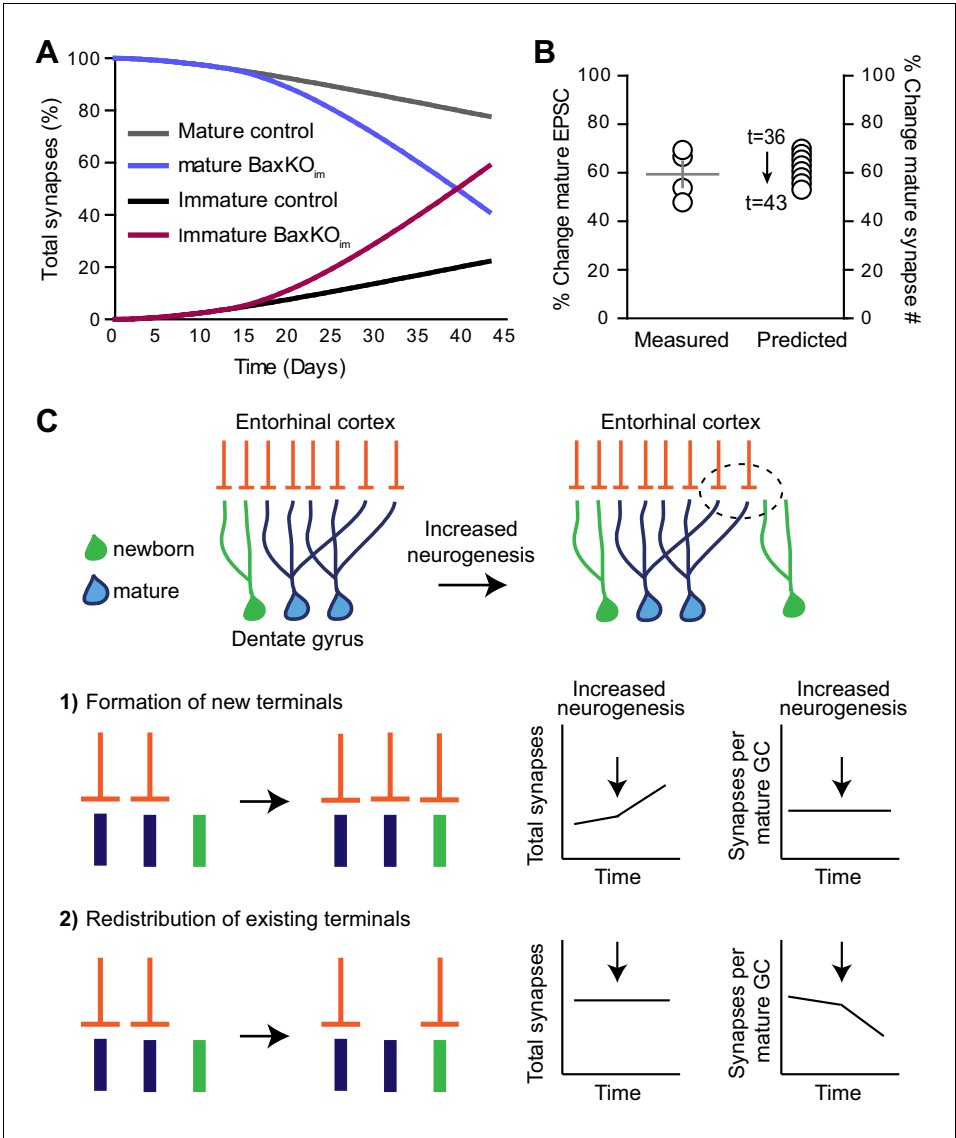

**Figure 9.** Simulation of neurogenesis-induced synaptic redistribution. (**A**) Distribution of synapses occupied by mature and immature GCs using quantitative synaptic transfer simulation (see Materials and methods). Lines indicate the percentage of synapses on mature and immature GCs across the duration of the BaxKO_{im} experiment, with the total number of synapses held constant. (**B**) Experimentally measured %change in EPSCs in mature GCs (left axis) compared to the %change in mature synapse number predicted by the simulation at time points t = 36 through t = 43 (right axis). Experimental data is the mean mature GC EPSC amplitude in BaxKO_{im} mice normalized to control from each FV bin shown in *Figure 1G*. (**C**) Graphic depiction of synaptic integration of adult born neurons showing that new GCs (green) gain EC synapses (orange) through two possible sources: (**C1**) New EC terminals may form to innervate new GCs. In this case, increasing neurogenesis would increase the total number of synapses over time but the synapses per individual mature GC would remain constant. (**C2**). Alternatively, new GCs may take over existing EC synapses from surrounding mature GCs. In this case, the total number of synapses would remain constant over time and the number of synapses per mature GC would decrease. The reduced synaptic input to mature GCs in BaxKO_{im} mice coupled with the apparent lack of change in total synapses (*Figures 1–3*) supports the synaptic redistribution model.

The following figure supplement is available for figure 9:

**Figure supplement 1.** Quantitative simulation of synaptic transfer.

results relate to prior anatomical studies. Nevertheless, our results unambiguously demonstrate that neurogenesis modifies synaptic transmission to existing mature GCs through a mechanism that involves reduced number of functional synapses. Unlike prior reports of alterations in DG excitability following selective manipulations of neurogenesis, we isolated excitatory synaptic transmission using $GABA_A$ receptor antagonists, thus our results cannot be attributed to differential recruitment of local inhibitory circuits by immature GCs (*Singer et al., 2011*; *Massa et al., 2011*; *Ikrar et al., 2013*; *Temprana et al., 2015*). In addition to such feedback inhibition, regulation of the density of mature GC excitatory synapses could potentially contribute to the counter-intuitive finding that the number of immature GCs is inversely related to the excitability of the mature network (*Ikrar et al., 2013*; *Drew et al., 2016*).

Our interpretation that integrating new GCs acquire synapses from mature GCs relies on the assumption that modulation of EPSCs reflects changes in synapse number. Several pieces of evidence support this assumption. First, to account for differences in the number of stimulated axons across slices, we normalized EPSCs in mature GCs to the simultaneously recorded fiber volley, a common approach used in synaptic plasticity studies. Thus, differences in EPSCs cannot result from systematic differences in the number of stimulated axons. Second, reduced evoked EPSCs were accompanied by reduced frequency of sEPSCs with no change in amplitude and no change in the PPR. These characteristics are widely accepted indicators of changes in synapse number. Third, strontium-evoked asynchronous EPSCs likewise supported the idea that small EPSCs in mature GCs from BaxKO$_{im}$ mice resulted from fewer active synapses rather than a postsynaptic change in sensitivity. We also found no difference in the $Ca^{2+}$ sensitivity of EPSCs between BaxKO$_{mat}$ and control mice. This suggests that *Bax* deletion from the majority of GCs did not affect $Ca^{2+}$ dependence of release processes, making it unlikely that a secreted factor acts presynaptically to alter release following *Bax* manipulation. Finally, we found that the density of mature GC spines was reduced after selective enhancement of neurogenesis. Our results are consistent with a model wherein newly generated GCs usurp pre-existing synapses from mature GCs, perhaps through an activity-dependent competitive process (*Tashiro et al., 2006*), yet we cannot rule out other non-competitive mechanisms by which newly generated cells affect the number of synapses on mature GCs. The recent observation that conditional suppression of spines on mature GCs enhances the integration of newborn GCs further supports the interactions between new and existing neurons (*McAvoy et al., 2016*).

A synaptic re-distribution model predicts that the addition of new neurons does not alter the total number of synapses within the circuit (*Figure 9C*). We used fEPSPs as a primary measure of total synapses, and presumably the fEPSP does not change despite the loss of EPSCs in mature GCs due to the additional contribution of synapses on immature neurons. Although we did not detect differences in fEPSPs (or vGluT expression), it is important to note that fEPSPs may not be particularly sensitive to synaptic density and will also be affected by intrinsic excitability. We did not detect any differences in the intrinsic excitability of mature GCs in our genetic models, but it is expected that the higher intrinsic excitability of immature neurons would enable a greater contribution to fEPSPs compared to mature GCs (for a given number of active AMPAR-containing synapses). However, newborn GCs have a high fraction of silent synapses that may limit their contribution to fEPSPs (*Chancey et al., 2013*). Most importantly, our interpretation of synaptic redistribution is not affected if the immature GC contribution to the fEPSP does not fully compensate for the loss of transmission to mature neurons (that is, *if the fEPSP was reduced in BaxKO$_{im}$ mice*). Only an *increase* in the fEPSP in BaxKO$_{im}$ mice would lend support a synaptic addition model. Even so, changes in fEPSPs are somewhat tangential to our novel finding that *EPSCs* in mature GCs are altered by selective manipulations of newborn GCs.

## Non-apoptotic role of the Bax signaling pathway in synaptic function

Our results indicate that Bax is required in mature GCs for neurogenesis-induced loss of transmission, suggesting that a change in the Bax signaling pathway is involved in spine loss from mature GCs. The contribution of Bax in our experiments is thus complex. We show that mature GCs exhibit a non-cell autonomous effect of *Bax* deletion from adult-born GCs (*Figures 1–3*, *decreased* EPSCs) that is opposite to the cell-autonomous effect of *Bax* deletion in both cell types (*Figures 5–6*, *increased* EPSCs). Remarkably, the cell autonomous function is required for the non-cell autonomous effect (*Figure 7*). This complexity, however, makes sense when we consider the role of Bax

signaling in both cell death and synapse pruning. We propose that the non-cell autonomous effect results from enhanced neurogenesis (supported by the observation that ablation of neurogenesis produced the opposite outcome, *Figure 4*), whereas the cell autonomous effect results from a contribution of the Bax pathway in synaptic depression and spine pruning.

Although the Bax signaling pathway is best known in the context of programmed cell death, it also has a non-apoptotic role in synaptic plasticity that is mediated by downstream caspases, the same family of cysteine proteases that initiate cell apoptosis (*Sheng and Ertürk, 2014*). Caspases mediate dendritic remodeling during neural development (*Kuo et al., 2006b*; *Williams et al., 2006*; *Riccomagno and Kolodkin, 2015*), and more recent work shows that caspase-3 activation is necessary and sufficient for NMDAR-mediated AMPA receptor internalization and LTD at hippocampal synapses (*Li et al., 2010*; *Jiao and Li, 2011*). LTD is associated with spine shrinkage and is typically considered a herald of synapse pruning (*Oh et al., 2013*; *Wiegert and Oertner, 2013*), thus it appears that pathways mediating cellular destruction also contribute to synaptic destruction (*Sheng and Ertürk, 2014*). Indeed, local induction of caspase-3 activity in dendrites triggers spine elimination whereas caspase-3 KO mice exhibit increased GC spine density (*Ertürk et al., 2014*; *Lo et al., 2015*), similar to our results of increased spine density in *Bax*KO GCs. Our findings that *Bax* deletion enhanced synaptic strength and spine density while blocking neurogenesis-induced loss of mature GC synaptic strength are consistent with the idea that on-going synaptic refinement controls the strength of excitatory transmission and that continual neurogenesis promotes a competitive environment for redistribution of synapses (*McAvoy et al., 2016*).

### Implications for dentate function

These results have potential implications for understanding the role of neurogenesis and plasticity in DG function. First, both enhancing neurogenesis and blocking output from mature GCs improves performance on the same context discrimination task (*Sahay et al., 2011*; *Nakashiba et al., 2012*), suggesting that neurogenesis could contribute to DG function by modifying mature GC activity. Synaptic depression and subsequent pruning are activity-dependent processes that typically require NMDA receptor activation (*Shipton and Paulsen, 2014*). Hence, re-distribution of active terminals away from mature GCs could transiently sparsify population activity, if new GCs initially have insufficient excitatory connectivity to allow recruitment (*Dieni et al., 2016*). Second, since eliminating *Bax* in progenitors leads to greater innervation as well as greater survival of neural progeny, enhancing neurogenesis by blocking the apoptotic pathway likely promotes competition to a greater extent than other methods of increasing neurogenesis. This could have implications for understanding the potential role of enhanced neurogenesis using *Bax* deletion in behavioral outcomes assessing pattern separation, stress resilience and forgetting (*Sahay et al., 2011*; *Akers et al., 2014*; *Hill et al., 2015*). Finally, our results showing that deletion of Bax signaling in postmitotic GCs enhances synaptic transmission is consistent with increased activation of DG neurons observed in caspase-3$^{-/-}$ mice, which also show behavioral deficits in attending to relevant stimuli (*Lo et al., 2015*). Together, we speculate that synaptic redistribution between immature and mature GCs may contribute to activity-dependent synaptic remodeling that allows salient stimuli to receive precedence in DG encoding and may also contribute to circuit remodeling that degrades established memories (*Weisz and Argibay, 2012*; *Chambers et al., 2004*; *Akers et al., 2014*; *Epp et al., 2016*).

## Materials and methods

### Transgenic mice

All animal procedures followed the Guide for the Care and Use of Laboratory Animals, U.S. Public Health Service, and were approved by the University of Alabama at Birmingham Institutional Animal Care and Use Committee (protocol# 8674 and 10134). Mice of either gender were maintained on a 12 hr light/dark cycle with *ad libitum* access to food and water.

BaxKO$_{immature}$ mice were generated by crossing heterozygous *loxP*-flanked *Bax* mice (Jackson #006329, the *Bak1* null allele was bred out) with Nestin-CreER$^{t2}$ mice (Jackson #016261). The offspring were crossed with each other to produce *Nestin-Cre$^+$* or $^-$/*Bax$^{fl/fl}$*, *Bax$^{fl/+}$*, or *Bax$^{+/+}$* animals (see *Figure 1—figure supplement 1*). Eight week-old mice were injected with tamoxifen (TMX, from a 20 mg/ml stock dissolved in sunflower seed oil, 75 mg/kg for three consecutive days) to induce

recombination and experiments were done 4–6 weeks post-injection. Control *Nestin-Cre⁻* or *Bax⁺/⁺* genotypes received TMX injections with the same protocol. For knockdown of neurogenesis, homozygous iDTR mice (Jackson #007900) were crossed with male Nestin-CreER^tm4 mice provided by Chay Kuo (*Kuo et al., 2006a*) to obtain offspring that were iDTR⁺ and either *Nestin-Cre⁺* (Ablated_immature) or *Nestin-Cre⁻* (control group). All mice were given TMX injections between 6–8 weeks of age, followed by diphtheria toxin injections six weeks later (DT, 16 µg/kg in sterile saline for three consecutive days). To conditionally delete Bax from postmitotic GCs, we crossed *POMC-Cre* mice (Jackson #010714) with *Bax^fl/fl* mice (see *Figure 6—figure supplement 1*). Conditional knockouts were maintained on a mixed 129 and C57BL/6J background using sibling controls. For counting newborn GCs, mice were crossed with *POMC-eGFP* transgenic mice (Jackson #009593). In some experiments, we visualized Cre-expressing cells by crossing conditional lines with Ai14 reporter mice (Jackson #007914). Tissue from homozygous germ line *Bax*KO mice (Jackson #002994) was used to validate Bax antibodies in western blots, with *Bax⁺/⁻* mice crossed with each other to generate both *Bax⁻/⁻* and control *Bax⁺/⁺* genotypes. All experiments were performed in adult P60-P120 mice.

## Electrophysiology

Mice were anesthetized and perfused intracardially with cold cutting solution containing (in mM): 110 choline chloride, 25 D-glucose, 2.5 $MgCl_2$, 2.5 KCl, 1.25 $Na_2PO_4$, 0.5 $CaCl_2$, 1.3 Na-ascorbate, 3 Na-pyruvate, and 25 $NaHCO_3$. The brain was removed and 300 µm horizontal slices were taken on a Vibratome 3000EP or Leica VT1200S (Leica Biosystems, Wetzlar, Germany). After recovery in artificial CSF (ACSF) containing (in mM): 125 NaCl, 2.5 KCl, 1.25 $NaH_2PO_4$, 2 $CaCl_2$, 1 $MgCl_2$, 25 $NaHCO_3$, and 25 glucose, recordings were performed at 30°C in ACSF +100 µm picrotoxin (PTX) to block $GABA_A$ receptors. Patch pipettes were filled with the following (in mM): 115 K-gluconate, 20 KCl, 4 $MgCl_2$, 10 HEPES, 4 Mg-ATP, 0.3 Na-GTP, 7 phosphocreatine, 0.1 EGTA, pH 7.2 and 290 mOsm (2–4 MΩ). In some cases, a 0.2% biocytin was included in the patch pipette. Field pipettes were placed in the middle molecular layer and filled with ACSF (1–2 MΩ). A patch pipette filled with 1M NaCl (1 MΩ) was used to stimulate the middle molecular layer using an isolated stimulator (Digitimer, Letchworth Garden City, UK). The minimum stimulation intensity that evoked an EPSC was first established and the stimulus intensity was increased at multiples of the threshold intensity until response saturation was evident. In some experiments we tested a pre-set range of stimulus intensities, again ceasing stimulation after responses saturated. Both methods used the same range of intensities (0 to 100 V) with each approach generating fewer independent observations at progressively higher stimulus intensities due to saturation of axonal recruitment. The response of 10 stimuli at each intensity was averaged. Averaged field EPSPs (fEPSPs) and EPSCs were binned by their corresponding fiber volley (FV) amplitude. This normalizes for differences in stimulus intensities across experiments and removes the parameter 'stimulus intensity' from data sets.

## Immunohistochemistry

Anesthetized mice were perfused intracardially with 0.9% NaCl or 0.1 M PBS and chilled 4% PFA before brains were removed and post-fixed overnight in PFA. Free-floating horizontal slices were taken on a Vibratome 1000 (50 µm). To enhance endogenous GFP expression, slices were blocked in TBS block buffer (0.1M TBS, glycine, 3% bovine serum albumin, 0.4% Triton X-100 and 10% normal goat serum) and incubated overnight with anti-GFP conjugated Alexa 488 (1:1000, Invitrogen, Carlsbad, CA). For NeuN and Dcx, slices were washed in TBST (50 mM Tris, 0.9% NaCl and 0.5% Triton X-100) and treated with antigen retrieval solution (10 mM sodium citrate, 0.5% tween 20) and 0.3% hydrogen peroxide before block with TBST +10% normal goat serum, followed by 48 hr incubation in rabbit anti-NeuN antibody (1:1000, Millipore, Billerica, MA) or rabbit anti-Dcx antibody (1:500, Abcam, Cambridge, UK), respectively. For NeuN, this was followed by incubation of 4 hr with goat anti-rabbit Alexa 647 (Invitrogen). For Dcx, a 3 hr incubation with biotinylated goat anti-rabbit (1:800, Southern Biotech, Homewood, AL) was followed by a 30 min incubation with streptavidin conjugated to Alexa Fluor 647 (1:200, Invitrogen). Slices were mounted with Prolong Gold or VectaShield mounting medium (Invitrogen). To visualize spines, acute brain slices containing biocytin-filled cells were post-fixed in 4% PFA for at least 24 hr then stained with streptavidin conjugated to Alexa Fluor 647 (1:1000, Invitrogen).

## Stereology

EGFP$^+$ cells and doublecortin (Dcx+) cells were counted using the optical fractionator method from every sixth slice through the entire left dentate gyrus using StereoInvestigator software (MBF Bioscience, Williston, VT). Counting frame and SRS grid sizes were set to give a Gunderson coefficient of error of <0.1 by an investigator blinded to genotype.

## Spine counting

For *Figure 3*, mature GCs from control and BaxKO$_{im}$ mice were patched using an internal solution that included 0.2% biocytin. After fixation, GC dendrites and spines were imaged on an Olympus Fluoview 300 confocal microscope with a 60X objective and a 3X digital zoom using a z-step of 0.1 μm. Dendritic segments that were relatively horizontal to the plane of the slice were selected for spine analysis by an investigator blinded to genotype (avg segment length = 46 ± 7 μm in control and 56 ± 4 μm in BaxKO$_{im}$ mice, p=0.3). Analysis of spine density and type was performed by an investigator blinded to genotype using NeuronStudio software (*Rodriguez et al., 2008*).

For *Figure 6*, TdTomato$^+$ (*Bax$^{-/-}$*) or tdTomato$^-$ (*Bax$^{+/+}$*) cells were patched in alternating slices from BaxKO$_{mature}$ mice and processed as described above. Spine density, length and head width were analyzed using Imaris software (Bitplane, Belfast, Northern Ireland)(*Swanger et al., 2011*).

## Bax protein analysis

Hippocampal lysates were prepared by homogenizing flash frozen subdissected hippocampi using RIPA buffer (150 mM NaCl, 50 mM Tris, pH 7.5, 1% Triton-X 100, 0.5% sodium deoxycholate, 1% sodium dodecyl sulfate) containing protease inhibitors (Fisher Scientific, Hampton, NH). Following BCA assay (Pierce), 20 μg of lysate was separated through 12% polyacrylamide gels and transferred to low-fluorescent PVDF (Biorad, Hercules, CA). Membranes were blocked with casein blocking buffer (Sigma-Aldrich, St. Louis, MO) in Tris buffered saline with 0.1% Tween 20 (TBST) and incubated with primary antibody (in 0.3% BSA in TBST) at 4°C overnight using antibodies to detect Bax (Fisher Scientific) or beta-tubulin (Developmental Studies Hybridoma Bank). Secondary antibodies conjugated to Alexa-680 (Fisher Scientific) allowed detection and quantification by scanning with an Odyssey Imaging System (Licor Biosciences, Lincoln, NE).

## Statistics

Data are expressed as mean ± SEM. We set the alpha level at 0.05 and accepted significant results with p<0.05 for all statistical tests. When determining the effect of genotype between two samples, data sets that satisfied normality criteria were analyzed with two-tailed paired or unpaired t tests, while non-normal data sets were analyzed with Mann-Whitney or Wilcoxon tests. For comparing two genotypes across multiple stimulus intensities, a two-way ANOVA was used. When EPSCs or fEPSPs were binned by FV amplitude, the number of data points varied between samples requiring an unweighted means analysis. Statistics were performed using Graphpad Prism.

## Quantitative estimate of synapse redistribution

The purpose of the calculation is to predict the proportion of mature GC synapses that will be appropriated by immature cells over a 6-week time period in a control or BaxKO$_{im}$ DG. Time (t) is expressed in days, where t = 0 represents the starting point when 8-week-old animals are injected with TMX. New GCs are continually added to an existing network comprised of mature and immature GCs. Each new GC gains synaptic strength beginning two weeks after cell birth (*Ge et al., 2006*; *Mongiat et al., 2009*; *Dieni et al., 2013*), acquiring innervation from a finite pool of synapses with synaptic strength defined as the number of synapses per cell. The total number of GCs was initially set at 200,000 (unilateral cell count in the adult mouse DG (*Pugh et al., 2011*). The number of mature GCs (>8 weeks cell age) was set at 95% of the total (190,000), while the initial number of immature cells (2–8 weeks cell age) was set at 5% of the total (10,000) (*Imayoshi et al., 2008*). The baseline number of mature GC synapses at t = 0 was set at 100%, defined as 100 per cell, giving initial mature synapse number, $S_M$:

$$S_M = 100(190,000 \times 0.95)$$

We approximated the increase in synaptic strength, $Y(t)$, of developing GCs by fitting the amplitude of evoked EPSCs in immature GCs at progressive ages (*Dieni et al., 2013*) by the equation:

$$Y(t) = 71.1ln(14+t) - 187.7$$

For example, a 2-week-old control GC receives ~5% as many excitatory synapses as a mature GC, a 5-week-old GC contains ~65% as many excitatory synapses, and an 8-week-old GC achieves 'mature' levels of 100% synaptic strength. To determine the initial number of immature synapses, $S_I(0)$, we divided the number of initial immature GCs by 43 (the number of days of maturation and thus the number of different synaptic strengths) and multiplied this quantity by the sum of all synaptic strengths:

$$S_I(0) = 10,000/43 \times \sum_{t=1}^{43} Y(t)$$

This result plus the initial number of mature synapses gives the total synapses in the system:

$$S_M + S_I(0)$$

which remains static throughout the simulation (~19.6 million).

To calculate the number of synapses appropriated by immature GCs each day, we considered cell proliferation $P(t)$, the rate of cell survival, and synaptic strength $Y(t)$. The rate of decrease in progenitor proliferation was defined by a best-fit equation (*Gil-Mohapel et al., 2013*), adjusted to give ~8000 progenitor cells at t = 14, (stereological ki67 counts from 8-week-old mouse) (*Chancey et al., 2013*), giving the available progenitor cell number, $P(t)$:

$$P(t) = 4 \times 10^6 (42+t)^{-1.5}$$

The survival rate for new WT cells is 20% (*Sierra et al., 2010*). In the BaxKO$_{im}$ group, new GCs incorporating into the network at t = 14 (2 weeks after TMX-induced recombination) have a survival rate of 70% (assuming partial efficiency of Cre expression)(*Lagace et al., 2007*). The number of immature GCs added to the system per day, $I(t)$, is:

$$I(t) = P(t) \times survival\ rate$$

All immature GCs will gain synaptic strength daily. The immature synapses appropriated each day, $S_I(t)$, is the cumulative sum of the surviving GCs times their respective synaptic strengths:

$$S_I(t) = (I(t) \times Y(1)) + (I(t-1) \times Y(2)) + (I(t-2) \times Y(3)) \ldots$$

Importantly, *Bax*KO GCs possess ~35% more synapses than control due to lack of Bax-dependent synapse pruning (*Figure 3E*, EPSC increase at highest FV bin).

In both groups, the cumulative number of immature synapses divided by the total synapses (multiplied by 100) equals the percent synapses appropriated by the immature population:

$$\%im = \frac{S_I(t)}{S_M + S_I(0)} \times 100$$

Since there is a static number of total synapses defined at the start of the simulation, the percent mature synapses remaining is:

$$\%mat = 100 - \%im$$

The %synapses occupied by all cell groups across time is plotted in *Figure 7*. Since the experiment is less than eight weeks in duration, immature GCs never convert into mature GCs, and we did not account for the conversion of pre-existing WT immature GCs because that population would not differ between control and BaxKO$_{im}$ conditions. To calculate the predicted difference in mature synapse number in BaxKO$_{im}$ vs. control conditions, we took the ratio of $\%mat$ in BaxKO$_{im}$ to $\%mat$ in control at each time point from t = 36 through t = 43 (multiplied by 100).

## Acknowledgements

We thank members of the Wadiche labs for helpful discussions throughout this project, Mary Seelig for technical assistance and Nancy Gallus for help with immunohistochemistry. This work was supported by Civitan International Emerging Scholars awards (EWA and HTV), F31NS098553 (RJV), NIH NS064025 (LOW), NIH NS065920 (JIW) and NIH P30 NS047466.

## Additional information

### Funding

| Funder | Grant reference number | Author |
|---|---|---|
| Civitan International | Emerging Scholars Award | Elena W Adlaf<br>Hai T Vo |
| National Institutes of Health | F31NS098553 | Ryan J Vaden |
| National Institutes of Health | P30 NS047466 | Gwendalyn D King<br>Jacques I Wadiche<br>Linda Overstreet-Wadiche |
| National Institutes of Health | NS065920 | Jacques I Wadiche |
| National Institutes of Health | NS064025 | Linda Overstreet-Wadiche |

The funders had no role in study design, data collection and interpretation, or the decision to submit the work for publication.

### Author contributions

EWA, CVD, Conception and design, Acquisition of data, Analysis and interpretation of data, Drafting or revising the article; RJV, AFM, Acquisition of data, Analysis and interpretation of data, Drafting or revising the article; AJN, MTA, Acquisition of data, Analysis and interpretation of data; VCO, Conception and design, Analysis and interpretation of data; HTV, Acquisition of data, Drafting or revising the article; GDK, JIW, LO-W, Conception and design, Analysis and interpretation of data, Drafting or revising the article

### Author ORCIDs

Gwendalyn D King, http://orcid.org/0000-0002-3659-9241

Jacques I Wadiche, http://orcid.org/0000-0001-8180-2061

Linda Overstreet-Wadiche, http://orcid.org/0000-0001-7367-5998

### Ethics

Animal experimentation: All animal procedures followed the Guide for the Care and Use of Laboratory Animals, U.S. Public Health Service, and were approved by the University of Alabama at Birmingham Institutional Animal Care and Use Committee (protocol# 8674 and 10134).

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
