## [Decision Letter]

Thank you for submitting your work entitled "Adult Born Neurons Modify Excitatory Synaptic Strength of Existing Neurons" for consideration by *eLife*. Your article has been reviewed by three peer reviewers, and the evaluation has been overseen by a Reviewing Editor and a Senior Editor. The reviewers have opted to remain anonymous.

Our decision has been reached after consultation between the reviewers. Based on these discussions and the individual reviews below, we regret to inform you that your work will not be considered further for publication in *eLife*.

Notably, each of the reviewers appreciated the potential impact of the paper, as little is known regarding the incorporation of newly born neurons into existing circuitry, and in particular, the potential compensatory adaptations of existing neurons that occur when new neurons are integrated. And the approach to increase or decrease neurogenesis and examine effects on excitability is sound. However each of the reviewers had distinct concerns, regarding validation of the neurogenic modifying approaches or electrophysiological analysis and results. While each of these could be addressed with additional experimental results it would not be possible to complete these within a reasonable time frame. As you are probably aware, *eLife* has a policy to not ask authors to do more than 2 months of work to return a revision, as we believe that asking authors to do considerable new experiments has slowed down the progress of science. Therefore, we are rejecting this manuscript now so that you are free to move on to another journal with the work in its present state should you wish. Of course, if you agree with the reviewers, you may wish to do the work suggested, in any case.

Reviewer #1:

The paper from Overstreet-Wadiche's lab seeks to understand the synaptic-level interrelationship between newborn GCs and mature GCs in the hippocampal dentate circuit. One view of in regards to the functional contribution of newborn GCs is that they possess unique intrinsic properties in the network, but an emerging view is that there are homeostatic adaptations within the network (e.g., feedback inhibition from adult-born GCs to mature GCs) that contributes to the circuit. This paper provides additional experimental support for the latter view, which is potentially significant in the field.

The main thesis of this manuscript is to understand how mature granule cells in the dentate gyrus respond to integrating adult-born GCs. They use two primary methods to manipulate endogenous neurogenesis; BaxKO in Nestin progenitors to boost neurogenesis and iDTR to ablate. They use a series of electrophysiological recordings in hippocampal slices to measure synaptic function. By stimulating afferent axons, they measured total synaptic function by means of fEPSPs and synaptic responses per GC by whole cell EPSCs under both conditions.

The two key findings are:

1) Bax deletion in newborn neurons enhance neurogenesis and results in a decreased synaptic transmission of mature GCs. This is demonstrated by decreased frequency of sEPSCs, but not amplitude, recorded from individual mature granule cells. In addition, using Sr+ to disrupt synchronous vesicular release, the data suggest a postsynaptic mechanism.

2) Conversely when NG is ablated synaptic transmission onto mGCs is increased. Similarly this is demonstrated by an increase in EPSC from whole cell recordings of mature granule cells.

They interpret these results with a model that suggest that as immature neurons start to integrate, the mature neurons reduce their synaptic strength in favor of the immature neurons, a so-called redistribution of pre-existing functional synapses.

To further support this provocative interpretation, they provide evidence that BaxKO new neurons have increased EPSC amplitudes compared to aged matched WT new neurons. And BaxKO in mature granule cells also showed larger amplitude EPSCs than neighboring BaxWT mature GCs. While this demonstrates that Bax plays a role in regulation synaptic function, the data is not yet convincing to support their major conclusion that the observed effects is due to a redistribution of pre-existing synapses. Also, there are some issues with their WT controls and validation of the Bax KO and iDTR genetic tools, which the authors could address with their existing resources.

Major Comments:

1) The generation of control for BaxKO appears to be littermates that lack NestinCreERT2 or floxed Bax gene. When the data is presented, it appears that both controls are combined, or at least not distinguished. It would be informative to separate out genotype controls to determine if the results are consistent with both controls to ensure there is not an effect specifically due to Cre expression.

2) The authors should show the number of newborn GCs reduced with their inducible DT-R model. Although this model has been previously published, there may be different efficiencies of newborn neuron ablation based on the number of cells that express DT-R after TAM administration and the length of DT treatment.

3) To compliment the functional data and bolster support for the redistribution hypothesis include immunohistochemistry (e.g. for AMPA receptors) of mGCs in BaxKO vs WT to anatomically quantify synaptic appropriation.

4) For Figure 5 and associated Supplementals, the data is comparing TdTom+ vs TdTom- or eGFP+ vs eGFP- and assuming the reporter is a direct indication of cre recombination in that cell. However, because it is independently expressed it is possible for the reporter to falsely identify recombination efficiency. How do you know TdTom+ cell has Bax deleted? Is there an antibody for Bax that could be counterstained with TdTom to show they do not colocalize?

Reviewer #2:

The paper by Adlaf and colleagues investigates how the incorporation of newly generated granule cells of the adult hippocampus alters excitatory connections onto preexisting granule cells. This is the other side of the coin that has not been looked at previously, focusing on how networks are remodeled to accommodate development and integration of new neurons. The authors used conditional Bax deletion to prevent apoptotic death of new neurons to increase neurogenesis, and found reduced excitation onto mature preexisting neurons due to a reduced number of synapses. When neurogenesis was reduced using diphtheria toxin, synaptic transmission onto mature neurons was increased, suggesting a competition for presynaptic terminals between new and old neurons. The authors also found that Bax deletion increases excitation in a cell-autonomous manner both in developing and in mature granule cells, and that synaptic competition requires Bax signaling. This is a very interesting work that approaches circuit remodeling by from an original perspective, necessary to understand in depth the consequences of neuronal addition in functional networks.

There are some experimental caveats that need to be addressed to strengthen the main conclusions in the manuscript.

1) In the experiments shown in Figure 1, the experimental design is such that Bax is specifically deleted in adult-born neurons, and the strength of synaptic transmission is decreased in old preexisting granule cells. Given that synaptic strength depends on the levels of Bax expression, it would be important to check whether Bax levels in old preexisting neurons are altered under these conditions that increase neurogenesis.

2) In Figure 2, the authors could look at spine density and see the structural correlates to functional changes. This is a simple experiment within the scope of the manuscript that would reinforce that changes in the number of synaptic connections are occurring.

3) In Figure 3, it is important to quantify the extent to which neurogenesis has been reduced by DT.

4) In the data shown in the Suppl data to Figure 3. no changes were found in neurotransmitter release probability, or freq of spont events or postsynaptic amplitude associated with the reduction in EPSC amplitude in iDTR mice. Thus, there are no parameters to explain the reduced strength. The authors state that it is unclear whether the reduction is pre or postsynaptic. However, this inconsistency between data obtained from evoked vs spontaneous release needs to be clarified experimentally or adequately explained (describing possible underlying causes).

5) Expt in Figure 6 show that only increasing neurogenesis is not sufficient to promote reduction of synaptic strength in preexisting neurons; Bax is required for this sort of competition (there is more neurogenesis, but in the absence of Bax in all neurons there is no difference in synaptic strength). Bax is required for what seems to be a neuron/neuron competition. Going back to the experiments shown in Figure 1, in this context it would be important to test whether increasing neurogenesis in a manner that is independent of Bax renders a similar reduction in EPSC strength. For instance, housing mice in enriched environment, that increases survival of adult-born neurons, should induce similar effects in old neurons. This is a very simple experiment.

6) Figure 4 Suppl also show no changes in parameters measured from spontaneous synaptic currents. Similarly to what was discussed in point 4, it would be expected that BaxKO neurons with more synapses show higher frequency of spontaneous synaptic currents. This discrepancy also needs clarification.

Reviewer #3:

The key finding of this study is that selective manipulation of adult born neurons is inversely correlated with the numbers of excitatory synapses on mature neurons, with no detectable changes in so-called global measures of synaptic transmission. The conclusion is that it is a re-appropriation of existing synaptic terminals that causes mature GCs to lose synapses while the total synapse number in the network remains constant. This is a potentially interesting finding with significant implications. However, there are several major concerns.

1) There is uncertainty concerning the basic finding in Figure 1. The fiber volley vs stimulus intensity plot in Figure 1 (left panel) shows clear saturation beyond 10x threshold, with the FVs reaching a plateau at around 100 microV even with a doubling of the stimulus intensity (compare 10x with 20x). Thus, stimulation beyond 10x clearly does not recruit any more afferent fibers. The saturation effect is also obvious in the fEPSP slope vs stimulation intensity plot (Figure 1, middle panel), where again the response reaches a plateau beyond 10x threshold. Therefore, it appears meaningless to use stimulation intensities beyond 10x threshold, or the corresponding FV value of 200 microV. But if we accept the latter premise, the argument that there is a reduced synaptic transmission to mature neurons without a change in fEPSP falls apart. To wit, in the fEPSP slope vs FV plot (Figure 1, right panel), the two BaxKOim data points at 100 and 200 microV are clearly (and probably significantly) smaller than their control counterparts. At these smaller stimulation intensities, the fEPSP vs FV and the EPSC vs FV plots actually look very similar. Note also that the largest difference in the EPSC amplitude vs FV plot in Figure 1 appears at the strongest stimulation intensity. Thus, the basic finding about the unchanged fEPSP in the face of altered single cell EPSC does not appear to be valid when responses to physiologically meaningful stimuli are considered.

2) Another fundamental methodological issue concerns the alleged lack of changes in PPR. The authors apparently used 100 ms ISI only, which invariably yields PPR values around 1 (Figure 2), meaning that at this ISI there was no short-term plasticity. Thus, the negative findings concerning PPR are not really interpretable from these data.

3) The authors only examined responses to stimulation of the middle molecular layer. Given that the study is concerned with global measures of synapse numbers in the dentate gyrus, other major excitatory inputs should be also examined and taken into account.

4) Why is it that the sEPSP frequency did not change in the immature neuron ablation experiment, whereas it did so when the number of immature neurons was enhanced?

5) A central conclusion of the study is that the induction of neurogenesis does not change the total number of synapses, and this argument is based on the (allegedly) unchanged fEPSP amplitude (but see above). Can one really use a relatively gross measure such as the fEPSP amplitude to argue for unchanged synapse numbers in a circuit, without direct anatomical assessment of synaptic density? This question is especially pertinent in a situation where EPSPs in different GC populations are expected to contribute to the field EPSP unevenly, since young and old GCs have different intrinsic properties.

6) The term synaptic strength is used in an ambiguous way in the text. Synaptic strength is usually used to define the efficacy of a given synapse, while here the proposed mechanism is a redistribution of functional synapses in the network. Receiving less or more inputs is not usually taken to mean that the synaptic strength changes.

7) What is the proposed mechanism of the hypothesized process of the transfer of synapses between immature and mature GCs?

---

## [Author Response]

*Notably, each of the reviewers appreciated the potential impact of the paper, as little is known regarding the incorporation of newly born neurons into existing circuitry, and in particular, the potential compensatory adaptations of existing neurons that occur when new neurons are integrated. And the approach to increase or decrease neurogenesis and examine effects on excitability is sound. However each of the reviewers had distinct concerns, regarding validation of the neurogenic modifying approaches or electrophysiological analysis and results. While each of these could be addressed with additional experimental results it would not be possible to complete these within a reasonable time frame. As you are probably aware, eLife has a policy to not ask authors to do more than 2 months of work to return a revision, as we believe that asking authors to do considerable new experiments has slowed down the progress of science. Therefore, we are rejecting this manuscript now so that you are free to move on to another journal with the work in its present state should you wish. Of course, if you agree with the reviewers, you may wish to do the work suggested, in any case.*

We thank the reviewers for constructive comments. We have added substantial new data, analysis and text revisions to address the concerns. In particular, we think the new data, summarized below, has strengthened our conclusions:

1) Analysis of FV/EPSC ratio in control groups (reviewer 1; new panels in Figure 1—figure supplement 1)

2) Quantification of iDTR ablation (reviewer 1 and 2; new panel in Figure 4)

3) Immunohistochemical detection of synapses (reviewer 1; new panels in Figure 4—figure supplement 1)

4) Spine density/type in mature neurons from BaxKOim and control mice (reviewers 1& 2; new Figure 3)

5) Assay of Bax expression in mature neurons (reviewer 2; new Figure 5—figure supplement 4)

6)Additional interstimulus intervals for the paired-pulse ratio (reviewer 3; new Figure 3)

7) Using environmental enrichment to enhance neurogenesis (reviewer 2; new Figure 8)

8)Data showing unchanged Ca^2+^ dependence of EPSCs (reviewer 3; new Figure 6—figure supplement 2)

We’d like to highlight a recently published article that likewise supports the idea of synaptic redistribution. McAvoy et al., Neuron (2016), show that synaptic integration of new neurons increases following conditional reduction of mature GC spines. We now show that selectively increasing neurogenesis decreases mature spine density, providing anatomical support for our main conclusion that adult born neurons modify synaptic connectivity of mature neurons.

*Reviewer #1:*

*[…]*

*Major Comments:*

*1) The generation of control for BaxKO appears to be littermates that lack NestinCreERT2 or floxed Bax gene. When the data is presented, it appears that both controls are combined, or at least not distinguished. It would be informative to separate out genotype controls to determine if the results are consistent with both controls to ensure there is not an effect specifically due to Cre expression.*

This is a good point. Since we recorded EPSCs primarily from mature neurons that do not ever express Cre, it seemed unlikely that Cre expression in newborn neurons affects our results. Nevertheless, to explicitly control for this possibility, we have now added new panels to Figure 1—figure supplement 1. Panel B shows there is no difference in the EPSC/FV ratio across control groups. We also included an additional analysis showing the EPSC/FV ratio difference persists using only mice with the Bax^fl/fl^ genotype, since Bax^fl/fl^ mice were originally obtained on a mixed B6;129 genetic background (panel C). We referenced these results in the text (Results section paragraph two). Finally, we did not detect differences in the synaptic properties of mature dentate GCs that underwent Cre-mediated recombination in POMC-Cre/Bax^wt/wt^ mice (Figure 6), further suggesting that transient Cre expression does not affect our results.

*2) The authors should show the number of newborn GCs reduced with their inducible DT-R model. Although this model has been previously published, there may be different efficiencies of newborn neuron ablation based on the number of cells that express DT-R after TAM administration and the length of DT treatment.*

We have now added stereological cell counts of Dcx-expressing immature neurons showing ~25% reduction in Dcx^+^ cells along with representative images (Figure 4). We included this data in the text (subsection “Ablation of immature neurons increases synaptic transmission to mature neurons”) along with the explanation that re-population of Dcx-expressing cells during the 1-2 weeks after DT-mediated ablation underestimates the ablation efficiency, as described in similar models (Yun et al., 2016).

*3) To compliment the functional data and bolster support for the redistribution hypothesis include immunohistochemistry (e.g. for AMPA receptors) of mGCs in BaxKO vs WT to anatomically quantify synaptic appropriation.*

We have now included data showing that expression of the presynaptic protein vGluT1 is unaltered in the molecular layer of BaxKO_immature_ and Ablated_immature_ slices (Figure 4—figure supplement 1, subsection “Ablation of immature neurons increases synaptic transmission to mature neurons”). However, this approach to label synapses by pre- or postsynaptic markers only assays total synapses, since it is not feasible to distinguish between mature or immature GC synapses.

Unfortunately, even using tdTom expression in immature neurons would not be sufficient to distinguish between mature and immature dendrites/spines because tdTom does not label the entire population of immature neurons.

To further address the request to anatomically quantify synaptic appropriation, we have now performed spine counts from mature GCs in BaxKO_im_ and control mice after whole cell recordings. We find that spine density of mature GCs is reduced, supporting the interpretation of our functional results. This data is included in new Figure 3 (with text in subsection “Enhancing immature neurons decreases EPSCs and spine density of mature neurons”).

*4) For Figure 5 and associated Supplementals, the data is comparing TdTom+ vs TdTom- or eGFP+ vs eGFP- and assuming the reporter is a direct indication of cre recombination in that cell. However, because it is independently expressed it is possible for the reporter to falsely identify recombination efficiency. How do you know TdTom+ cell has Bax deleted? Is there an antibody for Bax that could be counterstained with TdTom to show they do not colocalize?*

We agree that TdTom expression may overestimate recombination efficiency, and thus the difference in EPSCs shown in original Figure 5 (now Figure 6) underestimates the effect of Bax deletion. At the time we did the experiment, we were aware that Bax antibodies are notorious for poor labeling in brain slices and we subsequently confirmed this (see below). Thus we tested whether the Baxfl/fl allele was necessary for the difference in EPSCs in TdTom^+^ vs TdTom^-^ cells (new Figure 6) by performing the functional control experiment shown in Figure 6. This experiment shows that the Bax^fl/fl^ allele is necessary for the difference in EPSCs because repeating the experiment in TdTom^+^ Bax^wt/wt^ cells revealed no difference. We think this is a very strong control experiment, which also supports a lack of effect of transient Cre expression on synaptic function (see point 1, above).

To our knowledge, the only validated immunohistochemical identification of Bax in hippocampal slices shows weak diffuse labeling excluded from cell bodies using a non-fluorescent method (Figure 2 of Sun et al., J Neurosci 2004). Unfortunately we are unable to reliably detect Bax using this and other antibodies with fluorescence protocols, with comparison to tissue from germline BaxKO mice (Figure 10). This may be the reason why previous studies using conditional Bax deletion have not confirmed deletion with immunohistochemistry (Sahay et al., 2011; Ikrar et al., 2013; Hill et al., 2015). However, the increase in newborn neurons in BaxKO_immature_ and BaxKO_mature_ mouse lines identified by POMC-GFP expression independently verifies that Bax has been deleted from a significant fraction of newborn cells. In addition to the results shown in Figure 6, we found increased evoked EPSCs in 6 week-old and 16 week-old adult born conditional Bax-deleted GCs. Thus our data shows that EPSCs are enhanced in three separate experimental paradigms with Cre-mediated recombination in Bax^fl/fl^ cells, making it highly unlikely to be an experimental artifact.

Author response image 1.Bax immunostaining from WT slices (top) and germline BaxKO slices (bottom), using both standard (left) and antigen-retrieval methods (right).We are unaware of an antibody that reliably identifies Bax in hippocampal slices.
**DOI:**
http://dx.doi.org/10.7554/eLife.19886.027

*Reviewer #2:*

*[…]*

*There are some experimental caveats that need to be addressed to strengthen the main conclusions in the manuscript.*

*1) In the experiments shown in Figure 1, the experimental design is such that Bax is specifically deleted in adult-born neurons, and the strength of synaptic transmission is decreased in old preexisting granule cells. Given that synaptic strength depends on the levels of Bax expression, it would be important to check whether Bax levels in old preexisting neurons are altered under these conditions that increase neurogenesis.*

We did not intend to imply that synaptic strength depends of the level of Bax expression. Previous work shows that Bax is required for LTD because Bax activation is an intermediary step between NMDAR-Ca^2+^ influx and activation of caspase-3 that is necessary and sufficient for LTD (Li et al., Cell, 2010; Li and Jiao, Neuron 2011). Thus while Bax deletion prevents LTD (which leads to spine

shrinkage and eventual spine elimination (Nägerl et al., Neuron 2004; Zhou et al., Neuron 2004)), we are not aware of evidence that the level of Bax protein correlates with Bax activation or synaptic function/spine density. We have clarified the point that Bax activation is a key requirement in NMDAR- dependent hippocampal LTD in the text (subsection “Bax deletion in mature neurons increases EPSCs and spine density”).

Our results indicate that Bax is required in mature GCs for neurogenesis-induced loss of transmission (Figure 7), suggesting that a change in the Bax-caspase 3 signaling pathway is indeed involved in spine loss from mature GCs. Thus the contribution of Bax to our results is complex. We show that mature GCs exhibit a non-cell autonomous effect of Bax deletion from adult-born GCs (Figure 1–Figure 3, decreased EPSCs) that is opposite to the cell-autonomous effect of Bax deletion in both cell types (Figure 5–Figure 6, increased EPSCs). Remarkably, the cell autonomous function is required for the non-cell autonomous effect (Figure 7). This complexity makes sense when we consider the role of Bax-caspase 3 signaling in both cell death and synapse pruning (these points are now clarified in the Discussion).

We speculate that synapse loss from mature neurons involves enhanced Bax-caspase 3 activation rather than changes in Bax expression, as is reported for NMDAR-dependent LTD (Li and Jiao, 2011). We have now included western blot analysis showing that overall Bax protein levels are unchanged in BaxKO_im_ mice, confirming that deletion of Bax from a small percentage of cells does not change global levels of Bax protein (new Figure 5—figure supplement 4). In the absence of a reliable antibody for cell-type specific quantification in slices, this question cannot be more directly addressed with our currently available approaches. We are very interested in understanding this signaling pathway and its consequences for GC synaptic connectivity, but addressing this in more detail is likely beyond the scope of the current study.

*2) In Figure 2, the authors could look at spine density and see the structural correlates to functional changes. This is a simple experiment within the scope of the manuscript that would reinforce that changes in the number of synaptic connections are occurring.*

We thank the reviewer for this suggestion. We have now included spine analysis that shows reduced spine density in mature GCs from BaxKO_im_ mice (new Figure 3, text in subsection “Enhancing immature neurons decreases EPSCs and spine density of mature neurons”). This data confirms the interpretation of our functional analysis (Figure 1–Figure 2) of reduced number of synapses onto mature GCs in BaxKO_im_ mice.

*3) In Figure 3, it is important to quantify the extent to which neurogenesis has been reduced by DT.*

We have now added stereological cell counts of Dcx-expressing immature neurons showing >25% reduction in Dcx+ cells along with representative images (revised Figure 4). This data is included in the text (subsection “Ablation of immature neurons increases synaptic transmission to mature neurons”) along with the explanation that re-population of Dcx-expressing cells during the 1-2 weeks after DPT-mediated ablation underestimates the ablation efficiency (Yun et al., 2016).

*4) In the data shown in the Suppl data to Figure 3. no changes were found in neurotransmitter release probability, or freq of spont events or postsynaptic amplitude associated with the reduction in EPSC amplitude in iDTR mice. Thus, there are no parameters to explain the reduced strength. The authors state that it is unclear whether the reduction is pre or postsynaptic. However, this inconsistency between data obtained from evoked vs spontaneous release needs to be clarified experimentally or adequately explained (describing possible underlying causes).*

This is a good point that we now explain in the text. The frequency of spontaneous activity in mature dentate GCs is low (~1Hz), and primarily mediated by single site events since the amplitude is not affected by TTX (i.e. Dieni et al., J Neuro 2013, Parent et al., E J Neurosci 2016). Mature granule cells are estimated to have up to -5,000 synapses, such that spontaneous events represent the activity of only a small fraction of synapses, and there is high variability in the frequency of activity between individual GCs. Due to dendritic filtering and imperfect space clamp, it is also likely that spontaneous synaptic activity arising at proximal locations (i.e. synapses in the IML) are over represented, potentially contributing to “noise’ within and between experiments that does not occur in the evoked assay. While it is informative to see changes in spontaneous activity, it may be difficult to detect relatively small differences using this “lower resolution” assay, whereas evoked transmission repeatedly assays a higher fraction of synapses and there is less variability between cells after we use the FV to normalize the number of stimulated axons across slices. It appears that the change in neurogenesis in BaxKOim mice is greater than in iDTR mice (~40% vs ~25% change), and thus the “lower resolution” assay is unable to detect a difference in the iDTR model whereas the “higher resolution” assay can detect both. Finally, there is also ongoing debate about whether evoked and spontaneous synaptic release events actually arise from the same pool of vesicles, potentially underlying many reported differences in experimental results obtained by evoked and spontaneous activity (reviewed by Kavalali, Nat Rev Neurosci 2015). We have added this explanation in subsection “Ablation of immature neurons increases synaptic transmission to mature neurons”.

*5) Expt in Figure 6 show that only increasing neurogenesis is not sufficient to promote reduction of synaptic strength in preexisting neurons; Bax is required for this sort of competition (there is more neurogenesis, but in the absence of Bax in all neurons there is no difference in synaptic strength). Bax is required for what seems to be a neuron/neuron competition. Going back to the experiments shown in Figure 1, in this context it would be important to test whether increasing neurogenesis in a manner that is independent of Bax renders a similar reduction in EPSC strength. For instance, housing mice in enriched environment, that increases survival of adult-born neurons, should induce similar effects in old neurons. This is a very simple experiment.*

We have now included the requested experiment of increasing neurogenesis with EE/running (new Figure 8). We did not include this data in the original submission because this simple experiment requires a complex interpretation. Studies over the last 30 years have shown that EE increases synaptic connectivity in the DG, seen as increases in fEPSPs without changes in fiber volleys or paired-pulse ratios (Green and Greenough, JNeurophys 1986; Foster, Brain Res 1996) that can be explained by the increased spine density of mature GCs (Eadie et al., J Comp Neurol 2005; Stranahan et al., Hippocampus 2007, Glasper et al., Hippocampus 2010). We replicated the longstanding findings on fEPSPs and FVs (new Figure 8—figure supplement 1), and now directly show enhanced synaptic transmission to mature neurons with both evoked and spontaneous EPSC assays (new Figure 8). Our results are thus consistent with the prevailing evidence that EE increases synaptic connectivity of pre-existing mature GCs, in contrast to recent evidence that only adult born neurons in a critical period of development exhibit increased connectivity after EE (Bergami et al., Neuron 2015). Several differences in the cell populations tested or timing of EE could contribute to these differences. But altogether our results demonstrate the capacity of mature GCs to exhibit synaptic plasticity (in the form of altered connectivity) in response to both genetic and experiential circuit manipulations, a point that we now make clear in the Introduction.

Because EE increases the connectivity of mature neurons as well as the number of new neurons, this paradigm cannot resolve whether there is a redistribution of synapses. Increased synaptic connectivity (based on both anatomical and functional measures) induced by EE occurs in CA1 and other brain regions, indicating that it is a parallel mechanism independent of neurogenesis (Rampoon et al., Nat Neuro 2000; Stranahan et al., 2007; Glasper et al., 2010, De Bartolo et al., Brain Struct Funct, 2015; Bechard et al., Behav Brain Res 2016, etc). Thus EE could induce both outcomes (an increase in mature connectivity and an increase in new neuron synaptic redistribution), such that the “net effect” of EE on mature GC EPSCs shown in Figure 8 may be reduced by synaptic re-distribution to EE-induced new neurons. Extensive and additional experiments beyond the scope of this manuscript would be required to sort out this question.

We wholly agree that it would be useful to replicate our main result using an approach that is truly selective for enhancing the number of new neurons without affecting their integration or synaptic function, but we are unaware of an approach that has been fully validated in this regard. It seems that many proteins have dual roles in cell survival/cell cycle and synaptic function. Supporting this point, both proteins that were used to alter spine density in mature neurons in McAvoy et al., (2016; Klf9 and Rac1) may also be involved in apoptotic pathways (Stankiewicz and Linseman, Front Cell Neurosci 2014; Lebrun et al., Mol Cell Neurosci 2013) and our impression is that most genetic approaches that alter the number of new GCs also affect their maturation/dendrite structure.

While we cannot identify and validate a new approach to selectively increase neurogenesis without affecting integration, we point out that we used an alternative method to selectively reduce neurogenesis (Figure 4). In our view, demonstrating opposite manipulations of neurogenesis that lead to opposite changes in mature neuron EPSCs provides compelling evidence for our main conclusion that the number of new neurons affects synaptic connectivity of mature neurons.

*6) Figure 4 Suppl also show no changes in parameters measured from spontaneous synaptic currents. Similarly to what was discussed in point 4, it would be expected that BaxKO neurons with more synapses show higher frequency of spontaneous synaptic currents. This discrepancy also needs clarification.*

We reiterate that the low frequency of spontaneous activity makes it a less sensitive assay for changes in synaptic function than evoked activity. There are also many examples in the literature of discrepancies between evoked and spontaneous activity, which has led to ongoing debate about whether evoked and spontaneous synaptic release events arise from the same pool of vesicles (reviewed by Kavalali, Nat Rev Neurosci 2015). We have mentioned these points in the text. See also point 4 above.

*Reviewer #3:*

*The key finding of this study is that selective manipulation of adult born neurons is inversely correlated with the numbers of excitatory synapses on mature neurons, with no detectable changes in so-called global measures of synaptic transmission. The conclusion is that it is a re-appropriation of existing synaptic terminals that causes mature GCs to lose synapses while the total synapse number in the network remains constant. This is a potentially interesting finding with significant implications. However, there are several major concerns.*

*1) There is uncertainty concerning the basic finding in Figure 1. The fiber volley vs stimulus intensity plot in Figure 1 (left panel) shows clear saturation beyond 10x threshold, with the FVs reaching a plateau at around 100 microV even with a doubling of the stimulus intensity (compare 10x with 20x). Thus, stimulation beyond 10x clearly does not recruit any more afferent fibers. The saturation effect is also obvious in the fEPSP slope vs stimulation intensity plot (Figure 1, middle panel), where again the response reaches a plateau beyond 10x threshold. Therefore, it appears meaningless to use stimulation intensities beyond 10x threshold, or the corresponding FV value of 200 microV.*

We respectfully disagree that there is uncertainty in the result in Figure 1. This concern appears to be a misunderstanding of our experimental protocol and analysis, which likely resulted because we did not provide sufficient detail in the methods. We will explain the misunderstanding in detail, and describe how we have altered the figure and text to avoid this misunderstanding for other readers.

Author response image 2.Examples from six randomly selected experiments showing FV versus stimulus intensity relationships.Although the average graph (bottom right) suggests saturation of FVs around 200 µV (dotted line), the individual experiments reveal high variability in maximum FVs and saturation points (bottom left). We monitored saturation during each experiment and stopped stimulating when saturation was apparent (i.e. slice A,D,E).
**DOI:**
http://dx.doi.org/10.7554/eLife.19886.028

Reviewer 3 is correct that the averaged FV and fEPSP data shown in Figure 12 gives the appearance saturation. The averaged data, however, is not a good indicator of saturation in individual experiments because the stimulus-response relationships are highly variable between slices. We have illustrated this variability in a few examples of raw data and their average (Figure 11). The raw data shows variability both in the maximum FV amplitude (ordinate) as well as the stimulus intensity at which the maximum FV amplitude is achieved (the point of saturation on the abscissa). Hence the “linear range” of the stimulus response in each experiment occurs across a variable range of stimulus intensities. The goal of our stimulating protocol was to stimulate at multiple points within the linear range to acquire useful data across that range, but not above it. When saturation appeared (i.e. Slice A, D, E), we did not acquire responses at higher stimulus intensities. This results in variability in the stimulus intensities and intervals used (compare slice B versus E), and fewer data points in the averages for the highest range of stimulus intensities. Each individual response includes a range of FV amplitudes with a minimal number of saturated FVs. We have added the following description to the Methods to clarify how the experiments were performed:

“A patch pipette filled with 1M NaCl (1 MΩ) was used to stimulate the middle molecular layer using an isolated stimulator (Digitimer). The minimum stimulation intensity that evoked an EPSC was first established and the stimulus intensity was increased at multiples of the threshold intensity until response saturation was evident. In later experiments we tested a pre-set range of stimulus intensities, again ceasing stimulation after responses saturated. Both methods used the same range of intensities (0 to 100 V) with each approach generating fewer independent observations at progressively higher stimulus intensities due to saturation of axonal recruitment. The average response of 10 stimuli at each intensity was used for analysis. Field EPSPs (fEPSPs) and EPSCs were binned by their corresponding fiber volley (FV) amplitude. This normalizes for differences in stimulus intensities across experiments and removes the parameter “stimulus intensity” from all data sets.”

Thus the reviewer’s conclusion that responses are saturated at 10x threshold or at FV values above 100-200 µV are due to an over-interpretation of the averaged data plots of stimulus intensity and our failure to explicitly explain these plots. Whether the number of recruited axons is limiting (or saturated) is best assessed by the right panel of Figure 12 (circled) that plots the fEPSP slope versus FV. This linear relationship indicates that the postsynaptic response (fEPSP) does not saturate well above FV values beyond 100-200 µV.

We included all three plots in Figure 12 as an attempt to illustrate that the linear fEPSP vs FV plot (shown in most of our figures) results from the transformation of the two “messy” stimulus intensity plots (FV vs Stim and fEPEP vs Stim), revealing the linear (non-saturated) fEPSP-FV relationship once the “Stim Intensity” parameter is removed. We apologize if this presentation caused confusion rather than clarifying how the circled graph was generated.

*But if we accept the latter premise, the argument that there is a reduced synaptic transmission to mature neurons without a change in fEPSP falls apart.*

As shown in Figure 12 and described above, we did not include saturated responses in our analysis. The fEPSP versus FV relationship is linear.

Author response image 3.
**DOI:**
http://dx.doi.org/10.7554/eLife.19886.029

*To wit, in the fEPSP slope vs FV plot (Figure 1, right panel), the two BaxKOim data points at 100 and 200 microV are clearly (and probably significantly) smaller than their control counterparts.*

The fEPSP slope vs FV plot of slices from control and Baxim is not statistically significant (2way ANOVA). This outcome was replicated when we repeated the experiment (Figure 5—figure supplement 1; note this is the same experiment as shown in Figure 1, except that the mice include the Cre reporter TdTom and whole cell recording targets TdTom+ immature rather than unlabeled mature GCs).

*At these smaller stimulation intensities, the fEPSP vs FV and the EPSC vs FV plots actually look very similar. Note also that the largest difference in the EPSC amplitude vs FV plot in Figure 1 appears at the strongest stimulation intensity.*

The normalization procedure removes the parameter “stim intensity” from our analysis, leaving only the FV parameter that is a more robust measure of the number of axons recruited. Each FV bin contains data from a range of stimulus intensities (shown in lower left panel of Figure 11; it does not make sense to refer to stimulus intensities in binned FV data). Furthermore, most of our fEPSPs and EPSCs are in the smallest FV bins (now clarified in the figure legends) and all FV bins in Figure 1 are significantly different by Bonferonni post-tests (now added to legend).

*Thus, the basic finding about the unchanged fEPSP in the face of altered single cell EPSC does not appear to be valid when responses to physiologically meaningful stimuli are considered.*

Importantly, our interpretation about synaptic redistribution does not require that the fEPSP is unchanged, only that fEPSP does not increase (as one might initially predict from enhancing neurogenesis). Based on smaller EPSCs in mature GCs (Figure 1–Figure 3), one might indeed expect smaller fEPSPs since mature GCs make up the vast majority of GCs. Presumably the fEPSP does not change because of the additional contribution of immature synapses. Yet, our interpretation of synaptic redistribution does not require that the immature neuron contribution to the fEPSP fully compensates for the loss of transmission to mature neurons (that is, if we had actually found a reduction in the fEPSP, or if the sensitivity of extracellular recording is too low to detect such a difference). Only an increase in the fEPSP in Figure 1 would lend support a synaptic addition model (but even so, the decrease in mature EPSC and spine density still does not fit with an addition model). These are interesting and relevant points that we have now expanded on in the discussion, but the main point is that a smaller fEPSP does not affect our conclusions.

We reiterate that our central finding, that EPSCs in mature GCs are reduced in BaxKOim mice, is a robust result; the difference is apparent in the raw traces, it is accompanied by changes in frequency of spontaneous EPSCs and Sr2+ evoked EPSCs, and we now show that it is associated with reduced spine density of mature GCs (new Figure 3).

To avoid these misunderstanding by other readers, we have reformatted Figure 1 to make the presentation consistent with the other figures. We moved the averaged FV and fEPSP plots to Figure 1—figure supplement 2 and we provided more detail in the legends. We added a new panel to Figure 1 that shows the robustness of the difference in EPSC/FV ratios (independent of the FV amplitude) and we revised the methods to clarify that saturation was assessed in each individual experiment.

*2) Another fundamental methodological issue concerns the alleged lack of changes in PPR. The authors apparently used 100 ms ISI only, which invariably yields PPR values around 1 (Figure 2), meaning that at this ISI there was no short-term plasticity. Thus, the negative findings concerning PPR are not really interpretable from these data.*

We have now included additional ISIs for the paired pulse ratio experiment in new Figure 3. There was no difference in PPR at any ISI, as reported previously using fEPSPs in a similar model (S Figure 9 from Sahay et al., Nature 2011). We now cite Petersen et al., Neuroscience (2013) that shows MPP stimulation generates mildly depressing PPRs that depend, in part, on the stimulus intensity because it is difficult to isolate MPP from LPP stimulation (which displays PPF). We did not attempt to isolate the MPP rather we simply placed the stimulating electrode in the middle molecular layer, making it likely that our responses have some contamination from the LPP.

We are surprised by the assertion that a PPR near 1 is insensitive to changes in release probability, since manipulating release probability usually changes the PPR at short ISIs regardless of the initial PPR (i.e. Figure 8 of Debanne et al., 1996; Chancey et al., J Neuro 2014). To further address this concern, we now include data showing that the Ca^2+^ dependence of release is unaltered by Bax deletion from the majority of GCs (new Figure 6—figure supplement 2). This figure confirms that PPRs at 100 ms ISI are sensitive to changing the release probability and further rule out presynaptic changes following Bax deletion.

*3) The authors only examined responses to stimulation of the middle molecular layer. Given that the study is concerned with global measures of synapse numbers in the dentate gyrus, other major excitatory inputs should be also examined and taken into account.*

We reported changes in sEPSCs in Figure 1 that represent a global measure of synapse number rather than strictly medial perforant path synapses. While the results from additional examination of the A/C pathway or the lateral perforant path would be interesting and may perhaps help to define the extent of the change in mature neuron synaptic function, it is not clear that results from such experiments could alter our main finding since this request for additional data does not address a specific concern about our current data set.

*4) Why is it that the sEPSP frequency did not change in the immature neuron ablation experiment, whereas it did so when the number of immature neurons was enhanced?*

This is a good point that we now clarify in the text. The frequency of spontaneous EPSCs in mature dentate GCs is low (~1Hz), and primarily mediated by single site events since the amplitude is not affected by TTX (i.e. Dieni et al., J Neuro 2013, Parent et al., E J Neurosci 2016). Mature granule cells are estimated to have up to -5,000 synapses, such that spontaneous events represent the activity of only a small fraction of synapses, and there is high variability in the frequency of activity between individual GCs. Due to dendritic filtering and imperfect space clamp, it is also likely that spontaneous synaptic activity arising at proximal locations (i.e. synapses in the IML) are over represented, potentially contributing to “noise’ within and between experiments that does not occur in the evoked assay. While it is informative to see changes in spontaneous activity, it may be difficult to detect relatively small differences using this “lower resolution” assay, whereas evoked transmission repeatedly assays a higher fraction of synapses and there is less variability between cells after we use the FV to normalize the number of stimulated axons across slices. It appears that the change in neurogenesis in BaxKOim mice is greater than in iDTR mice (~40% vs ~25% change), and thus the “lower resolution” assay is unable to detect a difference in the iDTR model whereas the “higher resolution” assay can detect both. Finally, there is also ongoing debate about whether evoked and spontaneous synaptic release events actually arise from the same pool of vesicles, potentially underlying many reported differences in experimental results obtained by evoked and spontaneous activity (reviewed by Kavalali, Nat Rev Neurosci 2015). We have added this explanation in subsection “Ablation of immature neurons increases synaptic transmission to mature neurons”.

*5) A central conclusion of the study is that the induction of neurogenesis does not change the total number of synapses, and this argument is based on the (allegedly) unchanged fEPSP amplitude (but see above).*

Our central conclusion is indicated by the title (Adult-Born Neurons Modify Excitatory Synaptic Transmission to Existing Neurons) and in the last sentence of the Abstract:

“Together these results show that neurogenesis modifies the synaptic function of mature neurons in a manner consistent with a redistribution of pre-existing synapses to newly integrating neurons and that a non-apoptotic function of the Bax signaling pathway contributes to ongoing synaptic refinement within the DG circuit.”

As described in Point 1 above, we have a much softer interpretation of total synapse number than indicated by Reviewer 3 and smaller fEPSPs would not alter our conclusions.

*Can one really use a relatively gross measure such as the fEPSP amplitude to argue for unchanged synapse numbers in a circuit, without direct anatomical assessment of synaptic density? This question is especially pertinent in a situation where EPSPs in different GC populations are expected to contribute to the field EPSP unevenly, since young and old GCs have different intrinsic properties.*

We agree that the fEPSP may not be particularly sensitive to synaptic density and will be affected by intrinsic excitability (this is why we measured intrinsic excitability of mature GCs in all our protocols). We also agree that high intrinsic excitability will potentially allow immature GCs to have a greater contribution to fEPSPs (per AMPAR-containing synapse), potentially leading to similar fEPSPs despite smaller EPSCs in mature GCs.

Our new EE data includes a “positive control” that we can detect larger fEPSPs in a condition that is well known to increase the number of spines in the DG (replicating prior results of Green and Greenough, 1986 and Foster et al., 1996). We also cited prior work that found no change in anatomical assessment of synaptic density (measured with EM) despite a large increase in newborn cells in germline Bax knockout mice, and we have now also included an immunohistochemical assay suggesting presynaptic terminal density does not change (Figure 4—figure supplement 1). Even so, whether the fEPSP is smaller is tangential to our novel finding that EPSCs in mature GCs are altered by selective manipulations of newborn GCs.

*6) The term synaptic strength is used in an ambiguous way in the text. Synaptic strength is usually used to define the efficacy of a given synapse, while here the proposed mechanism is a redistribution of functional synapses in the network. Receiving less or more inputs is not usually taken to mean that the synaptic strength changes.*

We purposefully used the somewhat ambiguous term synaptic strength because it refers to a functional measure of synaptic transmission without assigning pre- or postsynaptic mechanisms to define it. Now that we have anatomical data that confirms the reduction in EPSCs in Figure 1 and Figure 2 are associated with fewer anatomically defined synapses, we have edited the text, including the title, to use more precise terminology.

*7) What is the proposed mechanism of the hypothesized process of the transfer of synapses between immature and mature GCs?*

Our data shows that deletion of Bax increases synaptic strength and spine density (new Figure 5) and that deletion of Bax in mature GCs blocks neurogenesis-induced loss of synaptic transmission (Figure 7). Since the Bax-/caspase-3 pathway is necessary and sufficient for AMPA receptor trafficking, NMDAR-dependent synaptic depression (LTD) and synapse pruning (references cited in the text), we speculate that the transfer of synapses involves synaptic depression (LTD) and subsequent pruning of mature GC synapses. We have clarified our discussion of this in the Results section and Discussion.